# PERCEPTUAL METRICS FOR VIDEO GAME PLAYSTYLE SIMILARITY AND DIVERSITY

## ABSTRACT

In gaming, decision-making diversity reflects the broad spectrum of styles that players can adopt. Despite the importance of this diversity, finding a universally applicable metric for it is challenging. To address this, a previous approach introduced the *Playstyle Distance*—a method for gauging similarity between datasets using game screens and their corresponding action pairs. This method identifies comparable states in discrete representations and then computes action distribution distances. Building on it, we introduce several new techniques. These include multiscale analysis with varied state granularity, perceptual kernels rooted in psychology, and the utilization of the intersection over union method for efficient data assessment. These innovations advance playstyle measurement and offer insights into human cognition of similarity. In experiments across two racing games and seven Atari games, our metric achieves over 90% accuracy in playstyle classification. Remarkably, this requires fewer than 512 observation-action pairs, less than half an episode in all tested games. We also develop an algorithm for assessing decision-making diversity using this metric. Our findings illuminate promising avenues for real-time game analysis and the evolution of AI with diverse playstyles.

## 1 INTRODUCTION

The pursuit of diversity in decision-making is one of the intrinsic motivations that drive human behavior, resulting in individual personality and creativity (Rheinberg, 2020). This is evident in the context of video games, where decision-making manifests as various playstyles, each reflecting individual strategies (Bean & Groth-Marnat, 2016). Alongside the pursuit of diversity, another central aspect of decision-making focuses on achieving optimal performance. Significant advances in decision-making performance have been witnessed, especially with the development of Deep Reinforcement Learning (DRL) (Russell & Norvig, 2020). The effectiveness of DRL was first showcased in arcade video games (Mnih et al., 2015). Subsequent applications to board games emphasized its potential, achieving superhuman skills (Silver et al., 2018). This success expanded into various types of games, from Agent57's superhuman agents in Atari games to groundbreaking feats in Dota 2 and StarCraft II (Badia et al., 2020; Berner et al., 2019; Vinyals et al., 2019). Beyond gaming, DRL applications extend to robot control (Lillicrap et al., 2016; Andrychowicz et al., 2020) and natural language processing (Ouyang et al., 2022), among others. These advancements in DRL underscore the profound impact of evolving decision-making techniques on modern technology.

Yet, while DRL continues to show promise in diverse applications, understanding and analyzing the playstyles it adopts remains a complex endeavor. Diverse data sources are essential for enhancing agent strength or efficiency (Fan & Xiao, 2022), just as diverse skill acquisition is vital for agents to generalize across tasks (Eysenbach et al., 2019). While a robust playstyle metric fosters a spectrum of playing strategies, it also reveals the inherent challenges in measuring these styles, particularly in environments without built-in features for playstyle measurement. Consequently, achieving precise playstyle measurement remains a formidable task (Tychsen & Canossa, 2008).

There are several methods to evaluate playstyles, from heuristic rules design to in-game features exploration (Tychsen & Canossa, 2008; Bontchev & Georgieva, 2018; Mader & Tassin, 2022). Supervised learning facilitates playstyle discrimination (Brombacher et al., 2017), while unsupervised clustering offers deeper behavioral insights (Ferguson et al., 2020). Another avenue involves con-

trastive learning to identify playstyles across samples (McIlroy-Young et al., 2021; Agarwal et al., 2021). Through these methods, the notion of playstyle can be gauged using distance or similarity metrics across game datasets, addressing the dynamic and evolving challenges in different scenarios. The concept is reflected in the work by Lupu et al. (2021), which specifies policy diversity using action distribution divergence but necessitates parametrized policies for similarity measurements.

The recent innovation by Lin et al. (2021) introduces the *Playstyle Distance* metric, which stands out by directly measuring playstyle from in-game observation-action pairs. Unlike common methods that compare latent feature distributions or rely on parametrized policies, this approach measures action distribution distances directly from gameplay samples, reducing the reliance on predefined playstyle data or extensive training sets for learning latent features or policies. Its effectiveness hinges on the critical role of state discretization. By discretizing observations, this method identifies comparable states, allowing a direct quantification of action distribution distances for each state, thereby encapsulating a player's style. Notably, the case-by-case estimation of states can withstand disturbances from uncontrollable variables, such as game randomness and interactions with other players. Consequently, it offers a consistent evaluation of playstyles across different games.

While the *Playstyle Distance* offers a pivotal advance, our research endeavors to elevate this foundation. We introduce innovative techniques to enrich the playstyle measurement paradigm, building upon *Playstyle Distance*. Initially, we leverage multiscale analysis with varied state granularity, emulating human judgment of similarity from multifaceted attributes and viewpoints (Medin et al., 1993). We then advocate for the use of perceptual kernels derived from psychophysics (Fechner, 1966) in psychology to achieve a probabilistic similarity value, harmonizing more with human comprehension than conventional distance values. Moreover, incorporating the *Jaccard index* concept (Murphy, 1996), we transcend the metric past mere intersections, fully harnessing the observed game dataset to tackle measurement unstability with sparse intersecting samples. These techniques not only improve precision in playstyle metrics but also shed light on understanding similarity through the lens of human cognition. From their fusion emerges the novel *Playstyle Similarity* metric.

To underscore our contributions, our metric's efficacy is validated across two racing games and seven Atari games. Remarkably, we attain over 90% accuracy in playstyle classification tasks with under 512 game screens observations and their corresponding action pairs—equating to less than half an episode in all examined games. In addition, we introduce a framework for measuring decision-making diversity, showcasing our metric's applicability. This comprehensive exploration magnifies its prowess in game analysis and AI training, harmoniously merging playstyle measurement with human-centric similarity interpretation.

## 2 BACKGROUND AND RELATED WORKS

In this section, we discuss playstyle metrics in-depth, provide a historical overview, and highlight the importance of discrete representation in crafting general playstyle metrics.

### 2.1 PLAYSTYLE AND MEASUREMENT

Establishing a universally accepted playstyle metric is a formidable challenge, as perceptions of playstyle are influenced by myriad factors and often harbor subjective nuances. Consequently, any playstyle metric should specify its evaluative parameters transparently to ensure its measurements are persuasive. Historically, tailored metrics, characterized by heuristic rules or specific in-game features, often presented the most precise for dedicated case studies. For instance, the study by Lample & Chaplot (2017) utilized metrics like object counts, kills, and deaths in shooting games. However, due to their inherent manual nature, these metrics are often domain-specific.

To achieve broader applicability, some researchers have resorted to supervised classification to identify target styles (Brombacher et al., 2017). However, this method requires labeled training data and may encounter difficulties in detecting styles that are not present in the training set. Unsupervised clustering offers a different angle, emphasizing latent feature distances for classification (Ferguson et al., 2020). But this approach may obscure the semantic meaning of the metrics, particularly when image data is the primary source. A noteworthy approach is the *Behavioral Stylometry* proposed by McIlroy-Young et al. (2021). This metric, crafted for chess playstyle assessment, encodes chess moves into a game vector, aggregates these vectors to represent a player, and then compares this

representation against a reference set. Central to this method is the contrastive learning technique, *Generalized End-to-End*, employed to differentiate players in training datasets (Wan et al., 2018).

For a more generalized measurement, one could consider measuring the similarity of action policies. It is common to regularize policy with a prior or given model (Schulman et al., 2015; Garg et al., 2023). Methods that extend to specify similarity or diversity by comparing the action distribution of two policies have also been explored (Agarwal et al., 2021; Lupu et al., 2021). Notably, these methods often require a parametrized policy for comparisons. This limitation is addressed by the *Playstyle Distance* metric (Lin et al., 2021). Instead of emphasizing latent features or parametrized policies, this method focuses on the action distributions of given samples. Observations are discretized and then leveraged for determining which action samples are comparable. Such a method resonates with human instinct more, echoing the case-by-case assessment we often deploy.

## 2.2 FRAMEWORK OF PLAYSTYLE DISTANCE

To delve deeper into the generality and importance of the *Playstyle Distance* metric in playstyle measurement, we examine its foundation as follows. A pivotal component of its methodology is the use of the Vector Quantised-Variational AutoEncoder (VQ-VAE), which specializes in discrete data representations by mapping continuous encodings to nearest vectors in a predefined codebook. This integration of neural networks with the nearest neighbor method results in discrete representations (van den Oord et al., 2017). Building upon VQ-VAE, the Hierarchical State Discretization (HSD) in *Playstyle Distance* ensures a concise and hierarchical state space. This is essential for pinpointing overlapping states while preserving the integrity of observation reconstruction and gameplay details.

Central to this framework is the discrete state encoder, denoted as $\phi$. In-game observations $o$ and their associated actions $a$ are mapped to datasets $M_i \sim Style_i$. The encoder $\phi$ translates these observations into a compact state representation $s$, formulated as: $S_\phi$: $\phi(o) \rightarrow s$. In the initial *Playstyle Distance* approach, the hierarchical encoder $\phi$ possesses the capability to generate multiple discrete states. However, the foundational literature employs a singular state space for computations. The reasoning for this choice and the potential to harness additional states are delved into in Section 3.

From state $s$, action distributions are deduced using a sampling distribution: $\{a|(o, a) \in M, \phi(o) \rightarrow s\} \Rightarrow \pi_M(s)$. Here, $\pi$ represents the policy, depicting action distributions for a given state. Subsequently, the distances between these distributions are determined using the metric $D(\pi_X, \pi_Y)$, where the 2-Wasserstein distance ($W_2$) serves as the standard (Vaserstein, 1969). Recognized for measuring the minimum 'effort' to transform one distribution into another, the Wasserstein distance is apt for policy comparisons, analogous to quantifying the 'effort' to transition between playstyles.

The essence of this metric can be succinctly captured as:

$$
\begin{aligned}
d_\phi(M_A, M_B) &= \frac{d_\phi(M_A|M_B)}{2} + \frac{d_\phi(M_B|M_A)}{2}, \\
\text{where} \quad d_\phi(M_X|M_Y) &= \mathbb{E}_{o \sim M_Y, \phi(o) \in \phi(M_X) \cap \phi(M_Y)}[D(\pi_X(\phi(o)), \pi_Y(\phi(o)))]
\end{aligned}
\tag{1}
$$

To encapsulate, the *Playstyle Distance* framework presents an advanced method to contrast gameplay styles, reducing the need for predefined heuristics and datasets, thereby increasing the applicability in various games. For a graphical representation of the framework, see Figure 1.

## 3 DISCRETE STATE PLAYSTYLE METRICS

In this section, we delve into a series of discrete state playstyle metrics derived from *Playstyle Distance*. We first address the limitations of *Playstyle Distance*. We then expand it into a multiscale approach by leveraging the hierarchical structure of states. Subsequently, we explore converting the action distribution distance into a perceptual value of similarity rooted in cognitive psychology, utilizing a perceptual kernel function. Thirdly, we broaden our scope from merely intersection states to the union of states, aiming for a more stable estimation of all observed data. Concluding the section, we integrate the strengths of these discrete state playstyle metrics into a comprehensive metric we term as *Playstyle Similarity*.

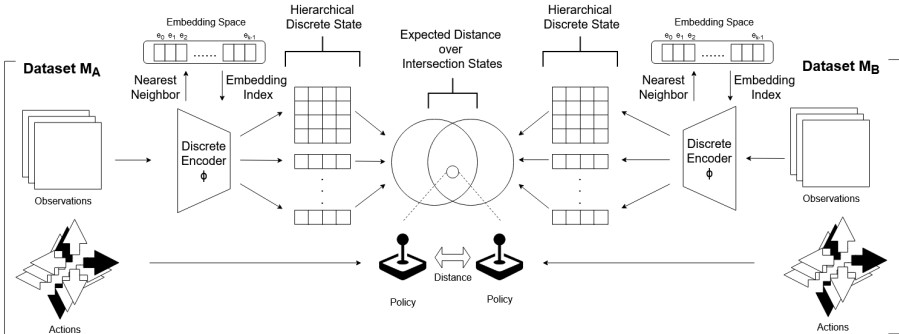

Figure 1: Illustration of the *Playstyle Distance* computation using a hierarchical discrete state encoder $\phi$. The Venn diagram highlights the intersection of discrete states for distance calculation.

## 3.1 EFFECT OF MULTISCALE STATES

*Playstyle Distance* underscores the importance of intersection states to ensure stable and precise distance measurements. Consequently, it resorts to a constrained state space sourced from the HSD model. A sample count threshold is applied to the intersection state, necessitating at least two samples in both datasets under comparison; failing which, the state is excluded from the intersection.

Human cognition perceives similarity as a convergence of multiple attributes leading to a holistic understanding (Goldstone & Barsalou, 1998). Hence, we advocate for employing varied granularity of discrete states to augment measurement capabilities, analogous to human judgment that varies from a broad view to intricate details. The HSD model's design inherently possesses a large state space for observational reconstruction and the discernment of gameplay nuances. Though the state space may be large, intersections are not void if observations are sufficiently similar or come from identical gameplay. It is worth noting that Lin et al. (2021) were able to distinguish intersection states even with unprocessed screen pixels in Atari games. In a different scenario, when treating each state as equivalent, we can invariably pinpoint an intersection state. In this context, distance simply gauges the action distribution over the entirety of the game, akin to traditional methods deploying post-game statistics. Broadly speaking, we can enhance the original state encoder function, $\phi$, evolving it into a state encoder mapping, $\Phi$, wherein $\Phi$ is an assemblage of mapping functions, $\phi \in \Phi$: $S_\Phi : \Phi(o) \to \{\phi(o) \in S_\phi | \phi \in \Phi\}$. Consequently, the projected state of dataset $M$ is defined as: $\Phi(M) \to \{\phi(o) \in S_\phi | o \in M, \phi \in \Phi\}$. We can then reinterpret Equation 1 as:

$$
\begin{aligned}
d_\Phi(M_A, M_B) &= \frac{d_\Phi(M_A|M_B)}{2} + \frac{d_\Phi(M_B|M_A)}{2}, \\
\text{where} \quad d_\Phi(M_X|M_Y) &= \mathbb{E}_{o \sim M_Y, \bigcup_{\phi \in \Phi}\{\phi(o) \in \phi(M_X) \cap \phi(M_Y)\}}[D(\pi_X(\phi(o)), \pi_Y(\phi(o)))]
\end{aligned}
\tag{2}
$$

This reformulated metric demonstrates superior accuracy in playstyle classification tasks in our experiments, even negating the need for a sample threshold count. This improvement is likely due to the metric's integration of hierarchical discrete states of varying granularity, which dilutes the impact of outliers during distance computation and leverages more useful details. Furthermore, the adoption of a multiscale state space effectively mitigates the trade-off between a compact intersection space and the preservation of intricate information details. This balance becomes crucial in complex games or those that require a vast state space to encode trajectory data.

## 3.2 PERCEPTION OF SIMILARITY

One potential shortcoming of the *Playstyle Distance* metric stems from the nature of distance itself. While distance is a common metric for determining similarity, a larger distance value conveys primarily that two entities are different, without giving much insight into the degree of their similarity.

Human cognition has been observed to exhibit the *Magnitude Effect*, suggesting diminished sensitivity to larger stimuli (Kahneman & Tversky, 1979). This is in line with the *Weber–Fechner Law* in psychophysics, where the relationship between stimulus and perception is logarithmic; as the

magnitude of stimuli increases, sensitivity diminishes (Fechner, 1966). Drawing from the concept of similarity, we can infer that a smaller distance provides more definitive information about the similarity. As distance grows, the distinction becomes vaguer. Therefore, we argue for a metric that reflects higher sensitivity to smaller distances, emulating human perceptual behavior.

We propose a probability-based model for similarity. In this model, greater similarity (i.e., smaller distance) corresponds to a probability closer to 100%, while lesser similarity (larger distance) approaches 0%. This proposed probability function aligns with the logarithmic human perceptual sensitivity to differences. Specifically, we use the exponential kernel to describe the probability of similarity, with the mapping function given by $P(d) = \frac{1}{e^d}$. This choice relates to the radial basis function kernel (Vert et al., 2004), and this perceptual kernel is the if and only if kernel under our assumptions from human cognition and probability. We provide a proof in Appendix A.1.

This exponential transformation can also be found in the Bhattacharyya distance and coefficient (Bhattacharyya, 1946). The Bhattacharyya coefficient $BC(P, Q)$ measures the similarity between two probability distributions $P$ and $Q$, and it is related to the overlapping region between these two distributions. It is defined as $BC(P, Q) = \int_{\mathcal{X}} \sqrt{P(x)Q(x)}dx$. The Bhattacharyya distance, derived from the coefficient, is $D_B(P, Q) = -ln(BC(P, Q))$, and the inversion is $BC(P, Q) = exp(-D_B(P, Q))$. Thus, a new playstyle metric $PS_\Phi^\cap(M_A, M_B)$ defined with probability of similarity based on Equation 2 is as follows:

$$
\begin{aligned}
PS_\Phi^\cap(M_A, M_B) &= \frac{\sum_{s \in \Phi(M_A) \cap \Phi(M_B)} P(D_\Phi^M(\pi_{M_A}(s), \pi_{M_B}(s)))}{|\Phi(M_A) \cap \Phi(M_B)|} \\
&= \frac{\sum_{s \in \bigcup_{\phi \in \Phi} \phi(M_A) \cap \phi(M_B)} P(D_\Phi^M(\pi_{M_A}(s), \pi_{M_B}(s)))}{|\bigcup_{\phi \in \Phi} \phi(M_A) \cap \phi(M_B)|}
\end{aligned}
\tag{3}
$$

$$
\begin{aligned}
D_\Phi^M(\pi_X, \pi_Y) &= \frac{D(\pi_X, \pi_Y)}{\overline{D}_\Phi^{M,X}}, \\
\text{where} \quad \overline{D}_\Phi^{M,X} &= \frac{\sum_{m \in M-X} \sum_{s \in \bigcup_{\phi \in \Phi} \phi(X) \cap \phi(m)} D(\pi_X, \pi_m)}{\sum_{m \in M-X} |\bigcup_{\phi \in \Phi} \phi(X) \cap \phi(m)|}
\end{aligned}
\tag{4}
$$

The metric has been simplified by adopting a uniform average distance instead of an expected value. This not only streamlines calculations but also underscores the significance of encoder granularity. In particular, an intricate encoder with a vast state space may be accorded greater weight, especially if the intricate encoder reveals more intersection states. Such a methodological tweak, while offering computational simplicity, doesn't compromise the quality of the outcomes, as evidenced by Lin et al. (2021). Moreover, by adjusting the distances with an average value, $\overline{D}_\Phi^{M,X}$, we ensure the expected distance converges to 1, consistent with our probabilistic framework (Appendix A.1). Collectively, our revamped metric provides a probabilistic lens to interpret similarity, firmly rooted in cognitive theory and tailored for human comprehension. There is more discussion about the role of the distance metric in Appendix A.2.1, including the implications of adopting different metrics.

### 3.3 BEYOND INTERSECTION

Before delving into our proposed final metric, it's pertinent to revisit the foundational concept of the *Playstyle Distance*: the intersection of states. The intuitive approach of identifying comparable states before gauging policy similarities runs into problems when the intersecting samples are limited. A smaller intersection could yield unstable or insufficient samples for measuring playstyles. Such a small intersection can be indicative of two scenarios. First, the distinct state-visiting distributions could signify different playstyles. Alternatively, uncontrollable factors, external to playstyles, may be involved, indicating the necessity for a more extensive sampling.

A prudent approach would assess the proportion of intersecting samples relative to the total observed samples. In the realm of collection comparison, the *Jaccard index* (Murphy, 1996), also known as *Intersection over Union*, emerges as a prevalent similarity metric. Notably, this method assumes no consideration of action distribution. The *Jaccard index* can serve as an effective playstyle metric under specific conditions. It is particularly apt when observational data clearly delineates playstyle

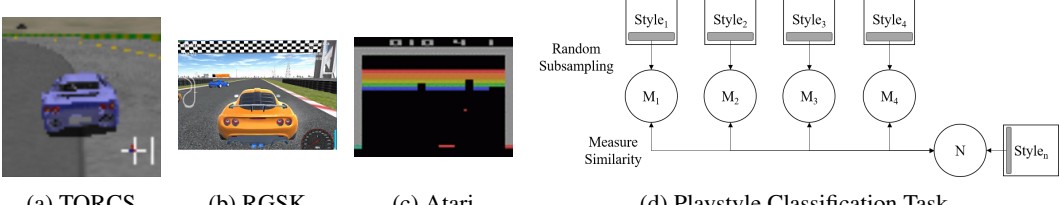

(a) TORCS  (b) RGSK  (c) Atari  (d) Playstyle Classification Task

Figure 2: Three game platforms and the illustration of playstyle classification tasks.

distinctions. For instance, in deterministic environments where states can be distinctly segmented by varying actions, the *Jaccard index* appears to be a fitting metric. However, complications arise when every state is visited or certain states recur due to game rules. The task of distinguishing different playstyles based solely on observations becomes considerably challenging. This is evident in single-state games, such as K-arm bandits (Sutton & Barto, 2018), where discerning playstyles based only on states becomes an impractical endeavor.

Despite potential challenges, our empirical findings suggest that the *Jaccard index* serves as a robust metric, especially when the state space is large and the randomness in game is low. The incorporation of the *Jaccard index* into a playstyle metric with a multiscale state space is expressed in Equation 5:

$$J_\Phi(M_A, M_B) = \frac{|\Phi(M_A) \cap \Phi(M_B)|}{|\Phi(M_A) \cup \Phi(M_B)|} = \frac{|\bigcup_{\phi \in \Phi} \phi(M_A) \cap \phi(M_B)|}{|\bigcup_{\phi \in \Phi} \phi(M_A) \cup \phi(M_B)|} \tag{5}$$

## 3.4 PLAYSTYLE SIMILARITY

Throughout our exploration, we have derived and discussed various playstyle metrics. Collating these insights, we introduce a comprehensive metric termed as the *Playstyle Similarity*. Defined as $PS_\Phi^\cup(M_A, M_B)$, it synthesizes our earlier discussions into a singular metric as illustrated below:

$$PS_\Phi^\cup(M_A, M_B) = J_\Phi(M_A, M_B) \times PS_\Phi^\cap(M_A, M_B)$$
$$= \frac{\sum_{s \in \Phi(M_A) \cap \Phi(M_B)} P(D_\Phi^M(\pi_{M_A}(s), \pi_{M_B}(s)))}{|\Phi(M_A) \cup \Phi(M_B)|} \tag{6}$$
$$= \frac{\sum_{s \in \bigcup_{\phi \in \Phi} \phi(M_A) \cap \phi(M_B)} P(D_\Phi^M(\pi_{M_A}(s), \pi_{M_B}(s)))}{|\bigcup_{\phi \in \Phi} \phi(M_A) \cup \phi(M_B)|}$$

What makes this metric novel is its unique treatment of intersection states. While the *Jaccard index* assigns a uniform weight (of 1) to each intersecting state regardless of the similarity between the action distributions, our approach infuses a more nuanced probability-based weighting. The values range between 0 and 1, increasing proportionally with similarity. This modification overcomes the potential limitation of using the *Jaccard index* for playstyle metrics.

Furthermore, our approach ensures a consistent interpretation of zero values. For states not part of the intersection, where the distance between action distributions is maximal (approaching infinity), they can be understood as entirely dissimilar.

## 4 EXPERIMENT SETTINGS

In this section, we explain the specifics of our experimental setup, focusing on the datasets, source of models, and our playstyle classification methodology.

### 4.1 GAME PLATFORMS, DATASETS, AND MODEL SOURCE

Our study encompasses three distinct game platforms, as depicted in Figure 2a, 2b, and 2c:

1. **TORCS**: This is a racing game with stable controlled rule-based AI players (Yoshida et al., 2017). The datasets derived from TORCS include a total of 25 playstyles based on driving speed and noise characteristics. Each observation consists of a sequence of 4 consecutive RGB images with a size of $64 \times 64$. The action space is 2-dimensional and continuous.

2. **RGSK - Racing Game Starter Kit**: This racing game, available on the Unity Asset Store (Juliani et al., 2020), showcases human players. From RGSK, we have data from a total of 24 players, exhibiting individual playstyles. Each observation from this game comprises 4 consecutive RGB images of size $72 \times 128$, with 27 discrete actions.

3. **Atari games with DRL agents**: The dataset spans 7 different Atari games (Bellemare et al., 2013) from this platform. Each game includes 20 AI models, all of which demonstrate varied playstyles. These AI models originate from the DRL framework, *Dopamine* (Castro et al., 2018). Each observation involves 4 consecutive grayscale images of resolution $84 \times 84$. The action space is discrete and varies depending on the game.

It is crucial to clarify that our research did not involve the training of new AI models. Instead, we leveraged three pretrained encoder models and corresponding datasets for each game, provided by Lin et al. (2021). The code and associated resources are available in their official release. [1]

## 4.2 PLAYSTYLE CLASSIFICATION AND STATE SPACE LEVELS

Our playstyle classification adheres to a specific methodology. As depicted in Figure 2d, we start with a target dataset $N$, sampled from a playstyle $Style_n$. We then compare this to multiple reference datasets $M$, each sampled from different playstyles $Style$. We perform 100 rounds of random subsampling for each playstyle; our primary performance metric for this task is the accuracy of playstyle predictions. If dataset $N$ exhibits the highest similarity to a reference dataset $M_i$, it suggests that $Style_n = Style_i$. It's worth noting that the reported accuracy represents an average, derived from results obtained using the three distinct encoder models.

Regarding the discrete state space levels, three tiers have been considered:

1. Space size 1, a basic mapping with state space 1, which maps all observations identically.

2. Space size $2^{20}$, as suggested by *Playstyle Distance*.

3. Space size $256^{64 \sim 144}$ or $256^{res}$, a level trained by HSD for the base hierarchy, depending on the resolution **res** of convolution features from game screens.

## 5 RESULTS AND DISCUSSION

In this section, we assess the efficacy of our proposed methods. Initially, we show that a multiscale state space can enhance the accuracy of *Playstyle Distance* without relying on a sample count threshold to preserve the quality of distance measurement. Subsequently, we contrast several baselines, illustrating that probabilistic methods for measuring similarity outperform distance-based approaches. Lastly, we incorporate all observed data to evaluate metrics across all platforms.

### 5.1 MULTISCALE STATE SPACE EFFICACY

To evaluate the efficacy of the proposed multiscale state space and to compare it fairly with *Playstyle Distance*, we primarily focus on the TORCS and RGSK platforms. Each sampled dataset from the given playstyles consists of 1024 observation-action pairs. Furthermore, we compare the sample threshold count $t$ of intersecting states. In this context, an intersecting state requires a minimum of $t$ samples in both datasets being compared for a more stable action distribution estimation.

We restrict our comparison to the multiscale version of *Playstyle Distance*, as defined in Equation 2. This equation is analogous to the original, as given by Equation 1, except that it uses only one discrete state space. For the multiscale variant which employs three discrete state spaces—$\{1, 2^{20}, 256^{res}\}$—we utilize the label **mix** for simplification. The results presented in Table 1 clearly indicate that employing a multiscale discrete state space not only delivers superior results but also obviates the need for a sample threshold count for intersecting states.

---

[1] https://paperswithcode.com/paper/an-unsupervised-video-game-playstyle-metric

|        | 1     | $2^{20}, t$=2 | $2^{20}, t$=1 | $256^{\text{res}}, t$=2 | $256^{\text{res}}, t$=1 | **mix**, $t$=2 | **mix**, $t$=1 |
|--------|-------|-------|-------|-------|-------|-------|-------|
| TORCS  | 35.16 | 70.68 | 60.83 | 4.60  | 60.76 | 74.77 | **75.05** |
| RGSK   | 80.14 | 78.39 | 93.43 | 5.62  | 26.58 | 88.51 | **94.21** |

Table 1: Playstyle accuracy (%) when employing various discrete state spaces in the multiscale version of *Playstyle Distance*, with a sample threshold count $t$ for intersecting states.

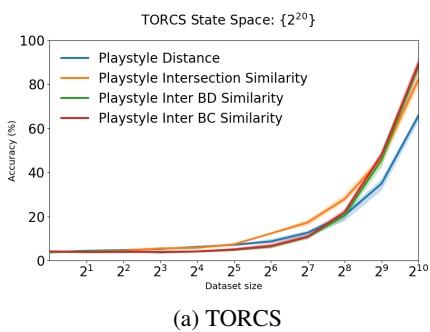
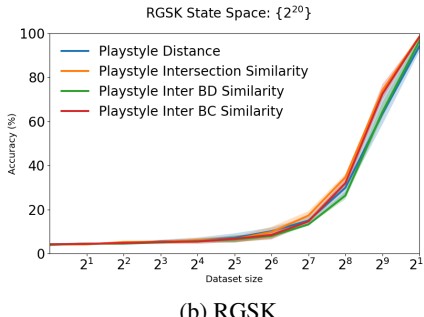

(a) TORCS             (b) RGSK

Figure 3: Comparison of Efficacy: Probabilistic vs. Distance Approaches. The plot illustrates the relationship between accuracy (Y-axis) and size of the sampled observation-action pairs (X-axis).

## 5.2 PROBABILISTIC VS. DISTANCE APPROACHES

In addition to introducing the multiscale discrete state space, another key contribution of our work is the proposal to use probabilistic similarity from a perceptual perspective rather than employing a negative distance as a measure of similarity. To elucidate the benefits of this modification, we study the relationship between accuracy and dataset size of the sampled observation-action pairs. These pairs are examined under a single discrete state space $\{2^{20}\}$, without employing a sample count threshold, to provide a clear evaluation of the transformation from distance to similarity. Further comparisons with different discrete state spaces can be found in Appendix A.2.

We evaluate several metrics in this comparison:

- *Playstyle Distance*: $-d_\Phi$
- Probabilistic similarity $PS_\Phi^\cap$, denoted as *Playstyle Intersection Similarity*.
- A variant of $PS_\Phi^\cap$ that employs the Bhattacharyya distance in place of the 2-Wasserstein distance, termed as *Playstyle Inter BD Similarity*, or $PS_\Phi^{\cap BD}$.
- The Bhattacharyya coefficient version, which omits the scaling coefficient before the perceptual kernel $\frac{1}{e^d}$, labeled as *Playstyle Inter BC Similarity*, or $PS_\Phi^{\cap BC}$.

Results presented in Figure 3 suggest that probabilistic similarity tends to yield more promising results compared to distance-based similarity. Among the methods, the 2-Wasserstein distance with perceptual kernel and Bhattacharyya coefficient are better candidates than distance similarity.

## 5.3 FULL DATA METRIC EVALUATION

In this section, we perform a comprehensive evaluation of various metrics, including leveraging full data with union operations. The evaluation methodology mirrors the one presented in Section 5.2, but widens the scope beyond just racing games and adopts a multiscale state space.

Given the page limitations of the main paper, detailed results for each Atari game have been moved to the Appendix A.3.1. Instead, leveraging the consistent observation and action space shared across Atari games, we propose a unified Atari console playstyle evaluation. This evaluation views each AI model's gameplay on individual games as distinct playstyles, yielding $7 \times 20$ unique playstyles. As for the discrete state space, we factor in a single shared state mapping in addition to two hierarchical discrete encoders from the seven games; thus, there are totally $1 + 7 \times 2$ discrete state encoders. For actions, rather than aligning their semantics across games, we simply expand the action set to the

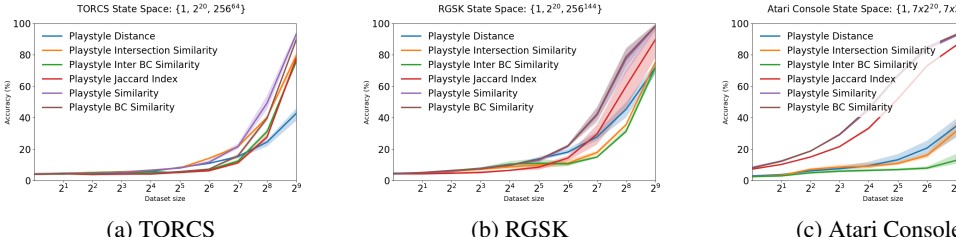

(a) TORCS        (b) RGSK        (c) Atari Console

Figure 4: Playstyle Metric Evaluation in TORCS, RGSK, and Atari Console. The plots showcase the efficacy of different metrics in the context of the "Full Data Metric Evaluation" subsection.

largest count found in Atari games, which is 18. This is based on the assumption that variations in game content can be interpreted as different states.

The platforms included in this experiment span TORCS, RGSK, and 7 Atari games. We're comparing the following metrics:

- *Playstyle Distance*: $-d_\Phi$
- $PS_\Phi^\cap$, termed *Playstyle Intersection Similarity*.
- $PS_\Phi^{\cap BC}$, termed *Playstyle Inter BC Similarity*.
- *Jaccard index*: $J_\Phi$, also referred to as the *Playstyle Jaccard Index*.
- $PS_\Phi^\cup$, or *Playstyle Similarity*.
- $PS_\Phi^{\cup BC}$, or *Playstyle BC Similarity*, the union version of *Playstyle Inter BC Similarity*.

Results displayed in Figure 4 show that the *Playstyle Similarity* outperforms its counterparts. Moreover, the *Jaccard index* has proven to be useful in practice. Our combined Atari console evaluation further underscores the robustness and adaptability of our metric.

Conclusively, our proposed *Playstyle Similarity* metric shines across all platforms. It's particularly impressive that it can identify playstyles with over 90% accuracy with just 512 observation-action pairs — less than half an episode across all tested games. This suggests the possibility of accurate playstyle prediction even before a game concludes, paving the way for real-time analysis.

## 6 CONCLUSION AND FUTURE WORK

In this research, we introduced three techniques to enhance playstyle metrics based on discrete states: adopting a multiscale state space, using perceptual similarity rooted in human cognition, and applying the intersection over union approach to observed data. These advancements have been incorporated into playstyle measurement for the first time and collectively give rise to our novel playstyle metric, *Playstyle Similarity*. This metric stands out in terms of accuracy and efficiency, requiring minimal predefined rules and data. Notably, the integration of a multiscale state space expands the metric's applicability, particularly for games that demand intricate state representations, such as trajectory information. Furthermore, our literature review and theoretical proof about human perception bridge the gap between AI-driven playstyle similarity metrics and human cognition.

The *Playstyle Similarity* metric offers significant potential for real-time game analysis and AI training tailored to specific playstyles, such as human-like behaviors (Fujii et al., 2013) or diversity measurement. As an example, we propose an algorithm to quantify of the diversity among DRL models with different stochasticities in Appendix A.4 by measuring the similarity between a new trajectory and observed trajectories. These insights emphasize that AI development should extend beyond simple measures like scores or win-loss ratios, encompassing nuanced behavioral patterns. Additionally, the exploration of diversity within decision-making models becomes more tangible.

In conclusion, although our assessments encompassed games such as TORCS, RGSK, and seven Atari games, a substantial portion of the gaming universe, spanning various genres and platforms, remains unexplored. Furthermore, some playstyles can be shared across different games, existing within semantically consistent game scenarios for game recommendation systems (Fear, 2023). Beyond the realm of gaming, playstyle metrics could be pivotal for topics related to decision-making, such as AI safety (Amodei et al., 2016), revealing vast potential for further exploration.

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

## A    APPENDIX

### A.1    A PROOF OF THE PERCEPTUAL KERNEL

In the main paper, we claim that $P(d) = \frac{1}{e^d}$ is the if and only if kernel function (as discussed in Section 3.2). We will provide a proof of this claim using differential equations.

We make some assumptions about the perceptual kernel. First, $P(d)$ is a function that maps the distance $d$ between two given action distributions to a probability value describing their similarity. Since distance is a continuous random variable, we use a probability density function $f(d)$ to describe the mapping function $P(d)$. We might intuitively think of $P(d)$ as equal to $f(d)$, even though the probability density value is not the same as the probability value.

Thus, we use a cumulative distribution function $F(d)$ to describe $P(d)$. We redefine a real-valued random variable $D$ as the distance variable, where $d \in D$, and a random variable $X$, where $x \in X$, and $D \to X : X(d) = -d$. The probability of similarity, denoted as $P(X \leq x)$, is derived from the distance value $d$. Thus, $F_X(x) = P(X \leq x)$, and from the assumptions of similarity, $\lim_{x \to -\infty} F_X(x) = 0$, and $F_X(0) = 1$. Also, $F_X(x)$ can be described with a probability density function $f_X(x)$ as follows:

$$F_X(x) = \int_{-\infty}^{x} f_X(t)dt \tag{7}$$

The corresponding equation to $F_D(d) = P(d)$ becomes:

$$F_D(d) = \int_{d}^{\infty} f_D(t)dt \tag{8}$$

Additionally, we adopt another assumption from the field of psychophysics known as the *Weber–Fechner law* Fechner (1966). Fechner's law states that the relationship between stimulus $S$ and perception $p$ is logarithmic and can be described as a differential equation:

$$dp = k\frac{dS}{S} \tag{9}$$

Here, $k$ is a constant depending on the sense and type of stimulus.

By integrating the equation, we obtain:

$$p = k\ln S + C \tag{10}$$

Where $C$ is a constant of integration, and it is defined in Fechner's law assuming that the perceived stimulus becomes zero at some threshold stimulus $S_0$, where $p = 0$, and $S = S_0$. Thus, $C$ can be calculated as follows:

$$C = -k\ln S_0 \tag{11}$$

Combining Equation 10 and Equation 11, Fechner's law Fechner (1966) is:

$$p = k\ln\frac{S}{S_0} \tag{12}$$

Now, we apply the roles of $p$ and $S$ in our similarity scenario to construct a probability density function $f_D(d)$. We assume that similarity weakens as the distance increases, and there is a finite maximal similarity when the distance is 0. Additionally, the density value and distance are always non-negative.

We first assume that $p$ is the density value of similarity and consider Equation 12. There are two cases:

1. If $S$ represents distance and $k$ is positive, this is incorrect since $p$ approaches $-\infty$ as $S \to 0$.

2. If we change the growth direction of distance so that $k$ is negative, this is still incorrect since $p$ still approaches $\infty$ as $S \to 0$.

Considering invert Equation 12 as follows:

$$S = S_0\exp\left(\frac{p}{k}\right) \tag{13}$$

Now, we assume that $S$ is the density value of similarity and consider Equation 13. There are two cases:

1. If $p$ represents distance and $k$ is positive, this is incorrect since $S$ approaches $\infty$ as $p \to \infty$, although there is a finite maximal value $S_0$ when $p = 0$.

2. If we change the growth direction of distance so that $k$ is negative, this seems to be true since $S$ approaches 0 as $p \to \infty$, there is a finite maximal value $S_0$ when $p = 0$, and the density value is always non-negative.

Finally, we can simplify the equations by assuming, for the sake of simplicity, that $S_0$ equals 1. This assumption is based on the intuition that the trend of decreasing similarity and increasing distance is similar around distance 0 in various scenarios:

$$f_D(d) = \exp\left(\frac{d}{k}\right) \tag{14}$$

Returning to Equation 8, $F_D(d)$ can be described as follows:

$$\begin{aligned}
F_D(d) &= \int_d^\infty f_D(t)dt \\
&= \int_d^\infty \exp\left(\frac{t}{k}\right)dt \\
&= \left(\lim_{t\to\infty} k\exp\left(\frac{t}{k}\right) + C'\right) - \left(k\exp\left(\frac{d}{k}\right) + C'\right) \\
&= -k\exp\left(\frac{d}{k}\right)
\end{aligned} \tag{15}$$

Considering that the sum of density values must be 1 over $(-\infty, \infty)$, we rewrite Equation 7 as follows:

$$\begin{aligned}
\lim_{x\to\infty} F_X(x) &= \int_{-\infty}^x f_X(t)dt \\
&= \int_{-\infty}^\infty f_X(t)dt \\
&= 1
\end{aligned} \tag{16}$$

The corresponding equation to $F_D(d) = P(d)$ becomes:

$$\begin{aligned}
\lim_{d\to-\infty} F_D(d) &= \int_d^\infty f_D(t)dt \\
&= \int_{-\infty}^\infty f_D(t)dt \\
&= \int_{-\infty}^0 f_D(t)dt + \int_0^\infty f_D(t)dt \\
&= 0 + \int_0^\infty f_D(t)dt \\
&= 1 \\
\implies F_D(0) &= 1
\end{aligned} \tag{17}$$

Combining Equation 15 and Equation 17:

$$\begin{aligned}
F_D(0) &= -k\exp\left(\frac{0}{k}\right) \\
&= -k \\
&= 1 \\
\implies k &= -1 \\
\implies F_D(d) &= \exp\left(\frac{d}{-1}\right) \\
\implies F_D(d) &= e^{\frac{1}{d}} \\
\implies P(d) &= e^{\frac{1}{d}}
\end{aligned} \tag{18}$$

Therefore, we have verified the claim that $P(d) = \frac{1}{e^d}$. If there is a case where $S_0 \neq 1$, it is straightforward to derive the equations from Equation 13 to 18.

Besides, the expected value of distance is 1 can be obtained by the equations as follows:

$$
\begin{aligned}
\mathbb{E}[D] &= \int_{-\infty}^{\infty} x f_D(x) dx \\
&= 0 + \int_0^{\infty} x f_D(x) dx \\
&= \int_0^{\infty} x \frac{1}{e^x} dx \\
&= (\lim_{t \to \infty} \frac{-t-1}{e^t} + C') - (\frac{-0-1}{e^0} + C') \\
&= 0 - (-1) \\
&= 1
\end{aligned}
\tag{19}
$$

This concept of expected value is used to scaling the distance value in different scenarios, as described in Section 3.2 with the notation $\overline{D}_{\Phi}^{M,X}$.

## A.2 PERCEPTUAL SIMILARITY UNDER DIFFERENT STATE SPACES

There are various methods for generating discrete representations, and the effectiveness of perceptual similarity may vary under these representations, especially when combined with our proposed multiscale state space. In this section, we explore the impact of different state space choices on perceptual similarity.

### A.2.1 BHATTACHARYYA DISTANCE IMPLEMENTATION

In this paper, we also provide some variants of *Playstyle Similarity*, which use Bhattacharyya distance or coefficient as an alternative to the 2-Wasserstein metric to assess the difference in playstyle from a different perspective. Bhattacharyya distance is related to the overlapping region between two distributions, and it is defined through Bhattacharyya coefficient $BC$. The value range of $BC$ is $[0, 1]$, and the corresponding distance $D_B$ is $D_B = -ln(BC)$. For discrete probability distribution, it is simple to compute the Bhattacharyya coefficient: $BC(P, Q) = \sum_{x \in \mathcal{X}} \sqrt{P(x)Q(x)}$. However, it is more challenging to calculate for continuous probability distributions, as in the case of actions in racing games like TORCS, since it involves the integration of probability density functions: $BC(P, Q) = \int_{x \in \mathcal{X}} \sqrt{p(x)q(x)}$. Thus, we adopt the formulation of multivariate normal distributions of Bhattacharyya distance ($D_B$) (Bhattacharyya, 1946) as follows, where $p_i = \mathcal{N}(\mu_i, \Sigma_i)$:

$$
\begin{aligned}
D_B(p_1, p_2) &= \frac{1}{8}(\mu_1 - \mu_2)^T \Sigma^{-1}(\mu_1 - \mu_2) + \frac{1}{2} \ln(\frac{\det\Sigma}{\sqrt{\det\Sigma_1 \det\Sigma_2}}) D_B, \\
\text{where} \quad \Sigma &= \frac{\Sigma_1 + \Sigma_2}{2}
\end{aligned}
\tag{20}
$$

Additionally, we clip the maximum Bhattacharyya distance to 10 to prevent an extremely large value from affecting the average scaling ($\frac{1}{e^{10}} = 0.00004539992 \approx 0\%$). The small value $\epsilon$ for dealing with singular matrices in matrix determinant calculation is set to 1e-8. For more detailed implementations, please refer to the code provided in our supplementary material.

Recalling our earlier discussion, we mentioned that the Wasserstein distance can be likened to the 'effort' required to transition between different playstyle action distributions (as described in Section 2.2). The Bhattacharyya distance, in contrast, isn't about this 'effort'. Instead, it gauges the likelihood that two playstyles will result in the same action. This is due to its relation to the overlapping regions between two distributions. Thus, while the *Playstyle Similarity* is built on the idea of the effort needed to change playstyles, the *Playstyle BC Similarity* (or its variant *Playstyle BD Similarity*) is built on the frequency of identical actions. This distinction might relate to different roles within a game. For instance, a player in the game might be more concerned with the effort required to shift playstyles, while an observer might focus more on the actions they witness. Think of it this way: players exert effort, like moving their fingers to press buttons or manipulate a joystick

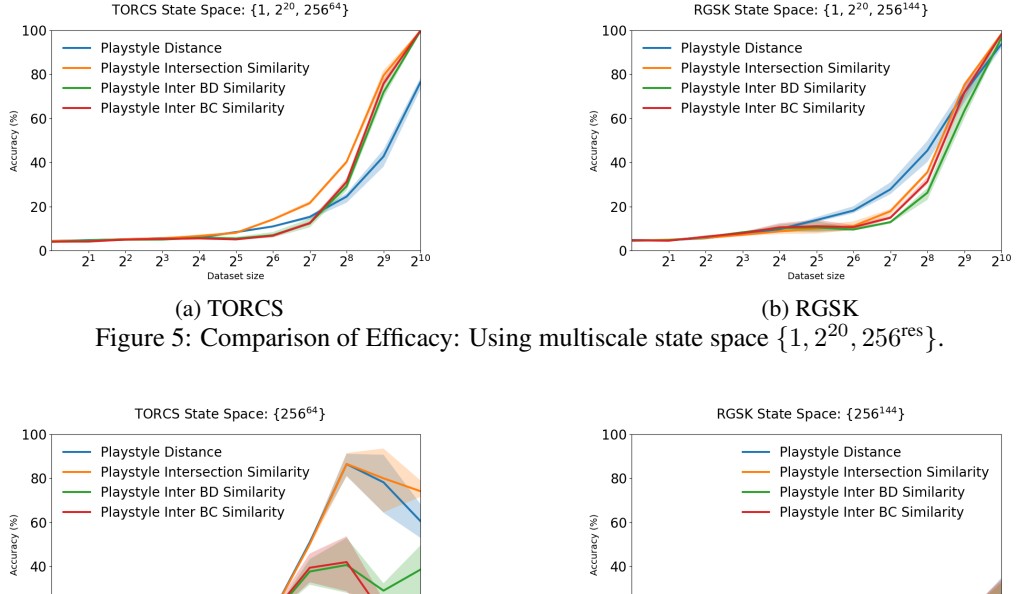

(a) TORCS        (b) RGSK

Figure 5: Comparison of Efficacy: Using multiscale state space $\{1, 2^{20}, 256^{\text{res}}\}$.

(a) TORCS        (b) RGSK

Figure 6: Comparison of Efficacy: Using only the base hierarchy of HSD with state space $\{256^{\text{res}}\}$.

or even a mental effort to change their belief of playing. The observer, on the other hand, sees only the outcome of these actions, without much insight into the effort involved.

### A.2.2 MULTISCALE STATE SPACE WITH HSD

Figure 5 presents the results of experiments conducted with a multiscale state space $\{1, 2^{20}, 256^{\text{res}}\}$ generated from HSD models, as described in Section 4.2.

The results indicate an improvement in accuracy for TORCS, while there is no clear improvement in RGSK. Notably, in RGSK, the accuracy of the perceptual kernel with sample sizes $2^5$ to $2^8$ decreases, suggesting that detailed information for distinguishing these styles has a negative effect. To further investigate, we conducted two ablation studies to understand the effectiveness of the proposed metrics for playstyle similarity. The first study focuses on using only the base hierarchy of HSD with a very large state space $\{256^{\text{res}}\}$, while the second study explores the use of a single-state state space $\{1\}$ to assess the metrics. Figure 6 illustrates that the measurement is unstable when there are few intersecting samples in a very large state space. However, the negative effect of detailed information is mitigated when considering intersection over union, as shown in Figure 4. Figure 7 shows that even with single state space, this action statistic of dataset can offer some information to differentiate playstyle, especially in RGSK, where Lin et al. (2021) made thier human players follow some playstyles closely related to the keyboard actions, such as using nitro system or breaking in racing games.

### A.2.3 DISCRETE REPRESENTATION FROM DOWNSAMPLING

There are several existing methods for generating discrete representations. One conventional method for image data is downsampling to a lower resolution. While the downsampling parameters often require tuning for effective processing, it is a straightforward method that does not require training a neural network model. Previous work by Lin et al. (2021) attempted to use low-resolution down-sampling as a discretization method, but they encountered challenges due to the lack of intersection states in their settings.

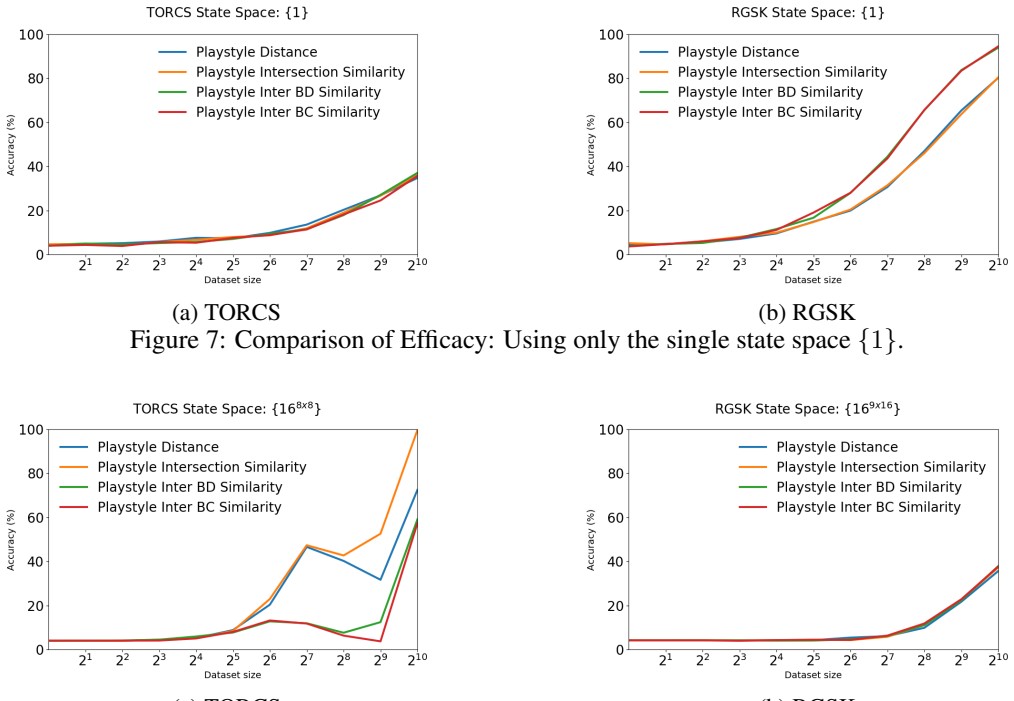

(a) TORCS          (b) RGSK

Figure 7: Comparison of Efficacy: Using only the single state space $\{1\}$.

(a) TORCS          (b) RGSK

Figure 8: Evaluation of discrete representations using downsampling, considering intersection states in TORCS with state space $\{16^{8\times8}\}$ and in RGSK with state space $\{16^{9\times16}\}$.

In our experiments, we explore the use of downsampling to create discrete representations with different state spaces. In TORCS, we map original game screen observations to three levels of state space:

1. 1: Basic mapping with state space 1, which maps all observations identically.

2. $16^{8\times8}$: Downsampling from $4 \times [64,64,3]$ 256-intensity observations to $1 \times [8,8,1]$ 16-intensity observations.

3. $16^{8\times8\times4}$: Downsampling from $4 \times [64,64,3]$ 256-intensity observations to $4 \times [8,8,1]$ 16-intensity observations.

For RGSK, we similarly map original game screen observations to three levels of state space:

1. 1: Basic mapping with state space 1, which maps all observations identically.

2. $16^{9\times16}$: Downsampling from $4 \times [72,128,3]$ 256-intensity observations to $1 \times [9,16,1]$ 16-intensity observations.

3. $16^{9\times16\times4}$: Downsampling from $4 \times [72,128,3]$ 256-intensity observations to $4 \times [9,16,1]$ 16-intensity observations.

The results in Figure 8-10 show that downsampling can be a viable discretization method in some cases, but overall, the measurement is either unstable or shows no significant difference compared in these metrics. These results highlight the importance of having discrete representations with high quality, providing proper granularity for playstyle features.

## A.3 More Results of Full Data Metric Evaluation

In this supplementary section, we delve deeper into the results to provide a comprehensive analysis of metric evaluations under different state spaces.

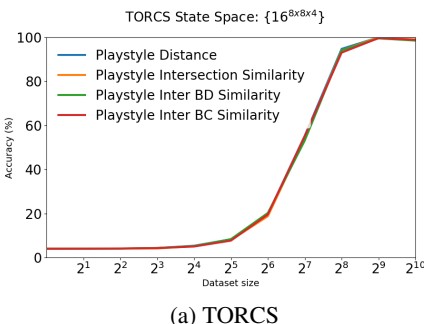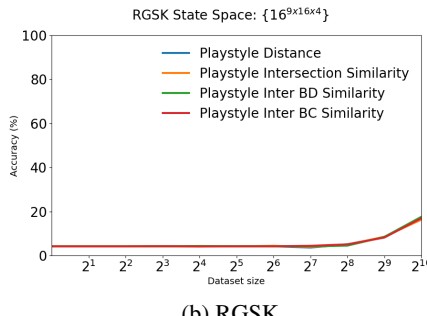

(a) TORCS

(b) RGSK

Figure 9: Evaluation of discrete representations using downsampling, considering intersection states in TORCS with state space $\{16^{8\times8\times4}\}$ and in RGSK with state space $\{16^{9\times16\times4}\}$.

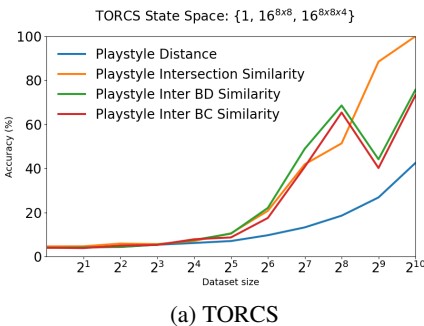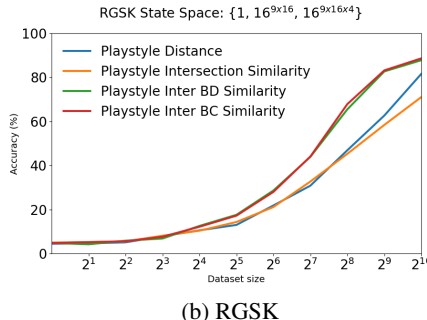

(a) TORCS

(b) RGSK

Figure 10: Evaluation of discrete representations using downsampling, considering intersection states in TORCS with state space $\{1, 16^{8\times8}, 16^{8\times8\times4}\}$ and in RGSK with state space $\{1, 16^{9\times16}, 16^{9\times16\times4}\}$.

### A.3.1 INDIVIDUAL ATARI GAME RESULTS WITH MULTISCALE STATE SPACE

Figure 11 shows the relationship between playstyle classification accuracy and sampled dataset size for the seven Atari games. *Playstyle Similarity* ($PS_\Phi^\cup$) and its variant *Playstyle BC Similarity* ($PS_\Phi^{\cup BC}$) have nearly the same performance, and *Playstyle Jaccard Index* ($J_\Phi$) can have a decent result. This evidence justifies that some playstyles, especially in a deterministic environment, can be differentiated solely with observations, which explains why the work by Eysenbach et al. (2019) considers states only for diversity.

### A.3.2 ATARI GAME RESULTS WITH A SMALLER STATE SPACE

Figure 12 shows the relationship between playstyle classification accuracy and sampled dataset size for the seven Atari games and the combined version (Atari Console). These results show that metrics with intersection over union still perform well in Atari games even with a smaller state space. Although it seems that *Playstyle Jaccard Index* is a decent and easy metric, we know that it theoretically does not work as long as all states are visited in the sampled dataset, as described in Section 3.3. This potential problem is discussed in more detail in Section A.3.3, where even with a state space of $2^{20}$, the *Playstyle Jaccard Index* may not perform well.

### A.3.3 TORCS AND RGSK WITH A SMALLER STATE SPACE

In this section, we conducted experiments using a reduced state space of $2^{20}$ for the two racing games, TORCS and RGSK, without employing the multiscale technique.

Figure 13 illustrates that the *Playstyle Jaccard Index* performs the poorest in TORCS and exhibits slightly inferior performance to *Playstyle Similarity* and *Playstyle BC Similarity* in RGSK. This observation provides valuable insights into the suitability of the *Playstyle Jaccard Index* for precise measurements, particularly in scenarios involving randomness (e.g., TORCS players employing different action noises) or where observations exhibit only slight variations (e.g., stable rule-based

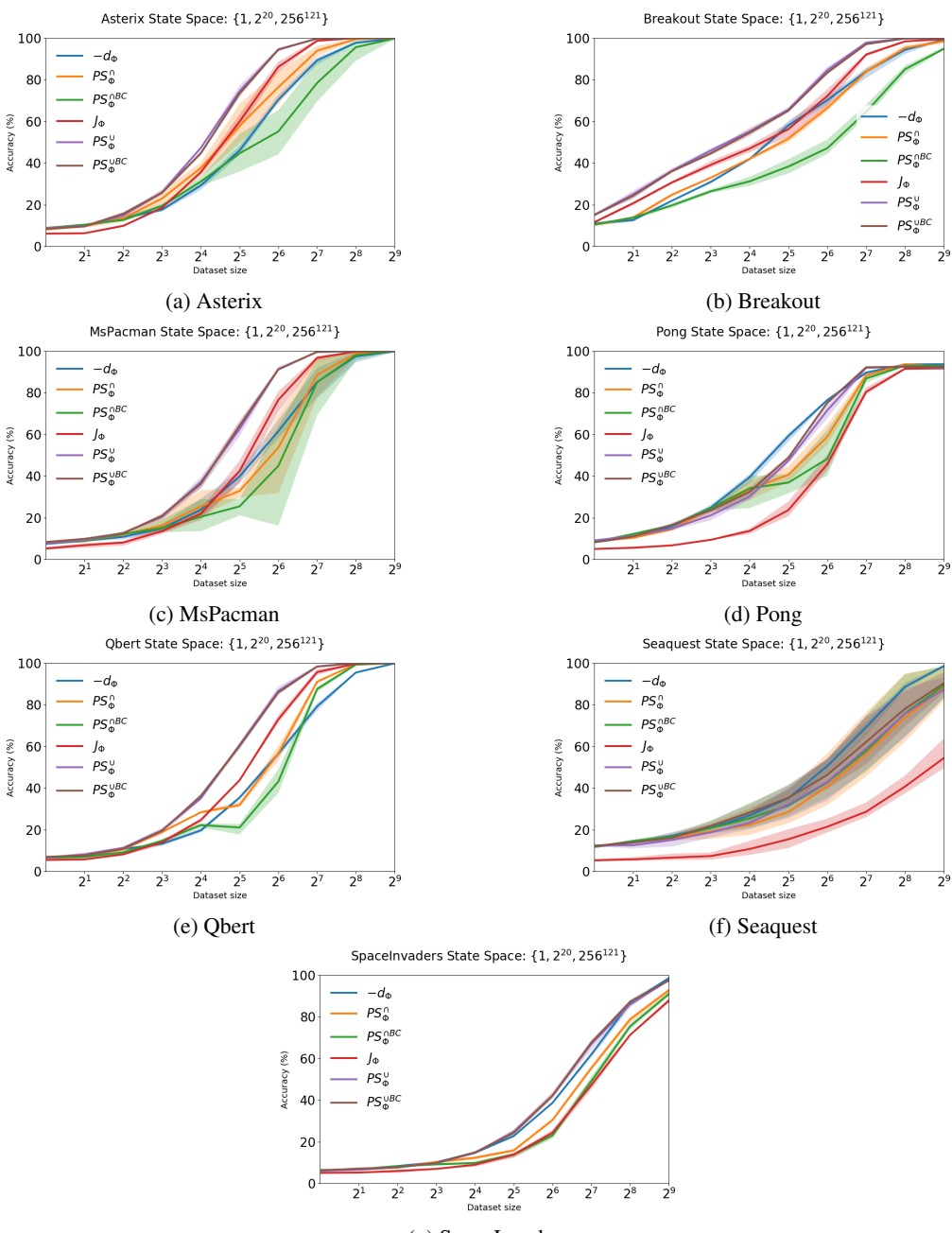

Figure 11: Playstyle Metric Evaluation in Atari games. The plots showcase the efficacy of different metrics in the context of the "Full Data Metric Evaluation" subsection.

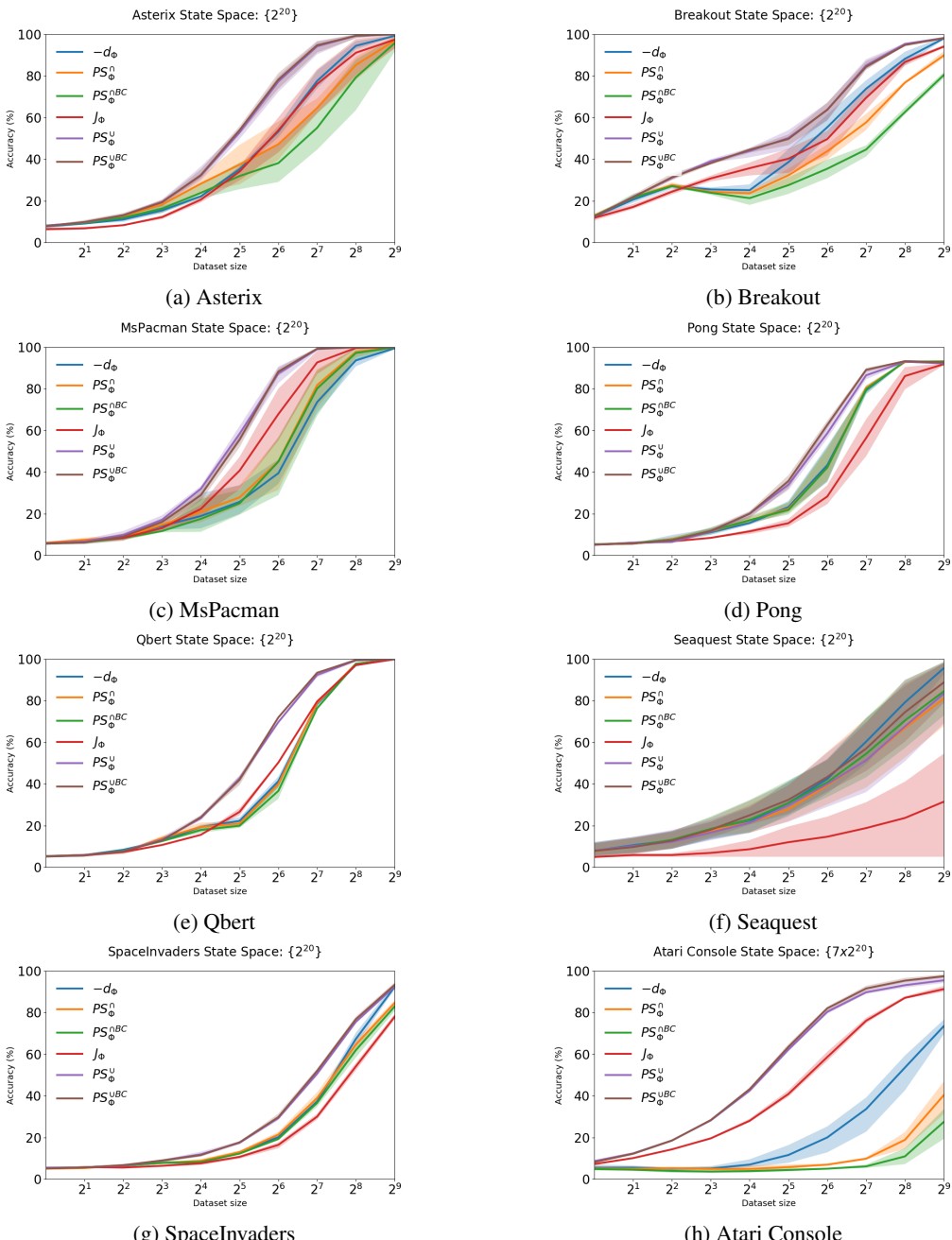

(a) Asterix

(b) Breakout

(c) MsPacman

(d) Pong

(e) Qbert

(f) Seaquest

(g) SpaceInvaders

(h) Atari Console

Figure 12: Playstyle Metric Evaluation in Atari games. The plots showcase the efficacy of different metrics with a $2^{20}$ state space from HSD models.

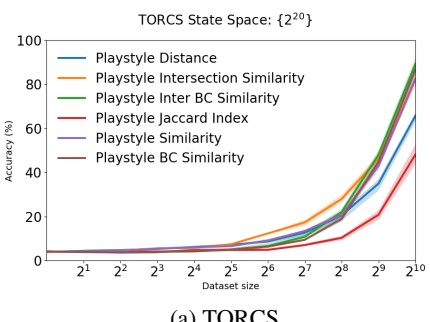 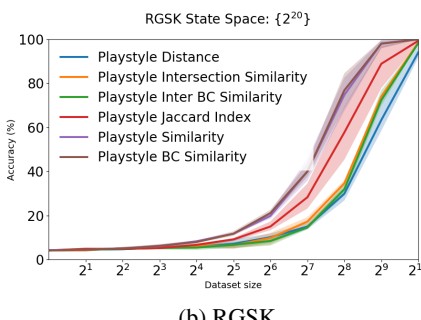

(a) TORCS                    (b) RGSK

Figure 13: Playstyle Metric Evaluation in two racing games. The plots showcase the efficacy of different metrics with a $2^{20}$ state space from HSD models.

AI controllers in TORCS with slightly different target). Further investigation may be warranted to understand the reasons behind these performance differences.

### A.4 DIVERSITY MEASUREMENT IN DRL

This section introduces a novel way of quantifying diversity in decision-making, detailed in Algorithm 1. We leverage models from the DRL framework, *Dopamine* (Castro et al., 2018), and apply various levels of stochasticity to illustrate this measure.

Our primary focus is on the first *IQN* (Dabney et al., 2018) model from *Dopamine*. This model should exhibit adaptability to a diverse array of playstyles, thanks to its risk functions and robust performance capabilities. Simply put, one could consider this model as akin to a proficient Atari player. To foster diversity, we use the Boltzmann distribution—a popular choice for stochastic categorical outputs, often referred to as the softmax distribution. By varying temperatures, denoted as $z$, and using $A$ to symbolize the advantage function for a given state $s$, the equation becomes:

$$\pi(s) = Softmax\left(\frac{A(s)}{z}\right) \tag{21}$$

This approach draws inspiration from the work by Fan & Xiao (2022). In reinforcement learning, the advantage function is a crucial value for selecting actions. An action with a higher advantage value is generally perceived as better.

---

**Algorithm 1** Measuring Policy Diversity

---

**Input:** Policy $\pi$, Environment $\mathcal{E}$, Similarity metric $M$
**Input:** Similarity threshold $t$, Number of trajectories $N$
  1: Initialize $S$ (store trajectories) and diverse trajectory count $d = 0$
  2: **for** $i = 1$ **to** $N$ **do**
  3:     Generate a trajectory $\tau_i \sim \pi, \mathcal{E}$
  4:     Set $is\_diverse =$ **true**
  5:     **for** each $\tau_j$ in $S$ **do**
  6:         Compute similarity $M(\tau_i, \tau_j)$
  7:         **if** $M(\tau_i, \tau_j) \geq t$ **then**
  8:             $is\_diverse =$ **false**
  9:             **break**
 10:         **end if**
 11:     **end for**
 12:     **if** $is\_diverse$ **then**
 13:         $d = d + 1$
 14:     **end if**
 15:     Store $\tau_i$ in $S$
 16: **end for**
**Output:** Return $d$ (diverse trajectory count) and $N$ (total trajectories)

---

Four levels of randomness are considered: $z \in \{0.0001, 0.001, 0.01, 0.1\}$. We expect diversity to increase with greater randomness. It is very simple to decide the similarity threshold $t$ for *Playstyle Similarity* metric, which can be linked to a probability value. Thus, we try $t = 0.5$ (50% similarity), $t = 0.2$ (20% similarity), and $t = 0.05$ (5% similarity) in seven Atari games with 100 trajectories each.

The discrete state spaces are defined in $\{1, 2^{20}, 256^{121}\}$, and the results from three HSD models are illustrated through shaded curves in the figures. The figures from Figure 14 to Figure 16 demonstrate the efficacy of our diversity measure.

Notably, games like *Seaquest* (Figure 17b) exhibit high diversity even at lower randomness levels, indicating intrinsic complexity in terms of playstyles. In contrast, *Qbert* (Figure 17a) becomes more monotonous when the goal is to achieve a higher score in the puzzle game. This observation suggests another application for our metric: identifying the complexity of game content. The time complexity of Algorithm 1 is $O(N^2)$, given the number of trajectories $N$. Future research could investigate more efficient methods, perhaps leveraging approximations or advanced data structures for quicker similarity checks.

The algorithm we introduce for diversity quantification shifts our understanding of gaming from the subjective to the quantitative. While various methodologies for measuring diversity exist in different domains, our approach is particularly apt to video game playing. In addition, recognizing and quantifying this diversity can inform the development of more adaptive DRL models, thereby addressing specific challenges in gaming and artificial intelligence. This new metric contributes to our progress toward models that are not only efficient but also demonstrate a variety of adaptable strategies, opening up vast avenues for future research.

## A.5  Uniform or Expected Weights for Discrete States in Playstyle Metrics

### A.5.1  Uniform Version of Playstyle Distance

$$
\begin{aligned}
d_\Phi(M_A, M_B) &= \frac{\sum_{s \in \Phi(M_A) \cap \Phi(M_B)} D\big(\pi_{M_A}(s), \pi_{M_B}(s)\big)}{|\Phi(M_A) \cap \Phi(M_B)|} \\
&= \frac{\sum_{s \in \bigcup_{\phi \in \Phi} \phi(M_A) \cap \phi(M_B)} D\big(\pi_{M_A}(s), \pi_{M_B}(s)\big)}{|\bigcup_{\phi \in \Phi} \phi(M_A) \cap \phi(M_B)|}
\end{aligned}
\tag{22}
$$

### A.5.2  Expected Version of Playstyle Intersection Similarity

$$
PS_\Phi^\cap(M_A, M_B) = \frac{PS_\Phi^\cap(M_A|M_B)}{2} + \frac{PS_\Phi^\cap(M_A|M_B)}{2},
$$
$$
\text{where} \quad PS_\Phi^\cap(M_X|M_Y) = \mathbb{E}_{o \sim M_Y, \bigcup_{\phi \in \Phi}\{\phi(o) \in \phi(M_X) \cap \phi(M_Y)\}}\big[P\big(D_\Phi^M(\pi_{M_X}(\phi(o)), \pi_{M_Y}(\phi(o)))\big)\big]
\tag{23}
$$

$$
D_\Phi^M(\pi_X, \pi_Y) = \frac{D(\pi_X, \pi_Y)}{\overline{D}_\Phi^{M,X}},
$$
$$
\text{where} \quad \overline{D}_\Phi^{M,X} = \mathbb{E}_{m \sim M - X}\Big[\frac{\mathbb{E}_{o \sim M_X, \bigcup_{\phi \in \Phi}\{\phi(o) \in \phi(M_X) \cap \phi(M_m)\}}\big[D(\pi_X(\phi(o)), \pi_m(\phi(o)))\big]}{2} +
$$
$$
\frac{\mathbb{E}_{\mathbb{E}_{o \sim M_m, \bigcup_{\phi \in \Phi}\{\phi(o) \in \phi(M_X) \cap \phi(M_m)\}}}\big[D(\pi_X(\phi(o)), \pi_m(\phi(o)))\big]}{2}\Big]
\tag{24}
$$

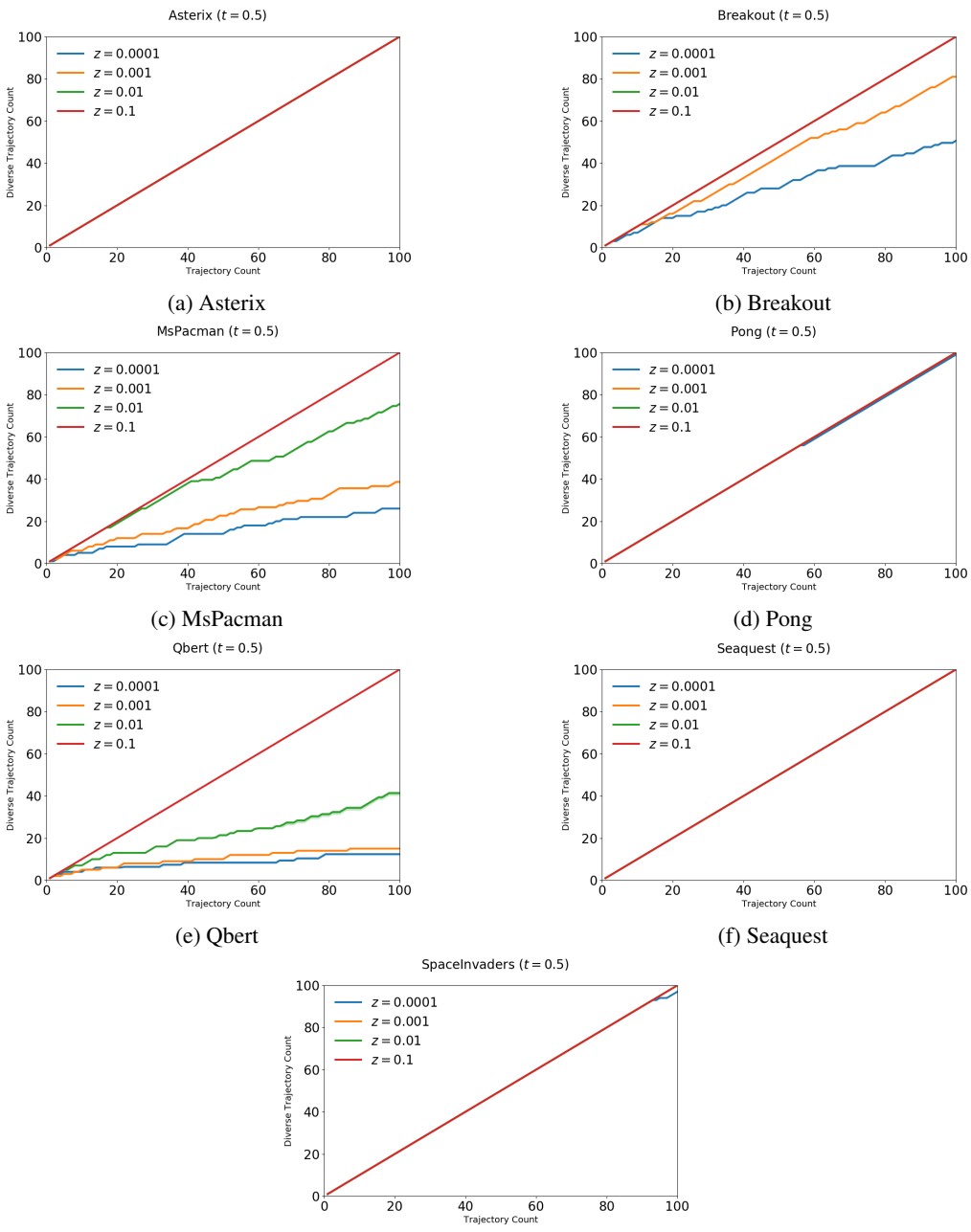

(a) Asterix

(b) Breakout

(c) MsPacman

(d) Pong

(e) Qbert

(f) Seaquest

(g) SpaceInvaders

Figure 14: Diversity Measurement of IQN Models in Seven Atari Games ($t = 0.5$)

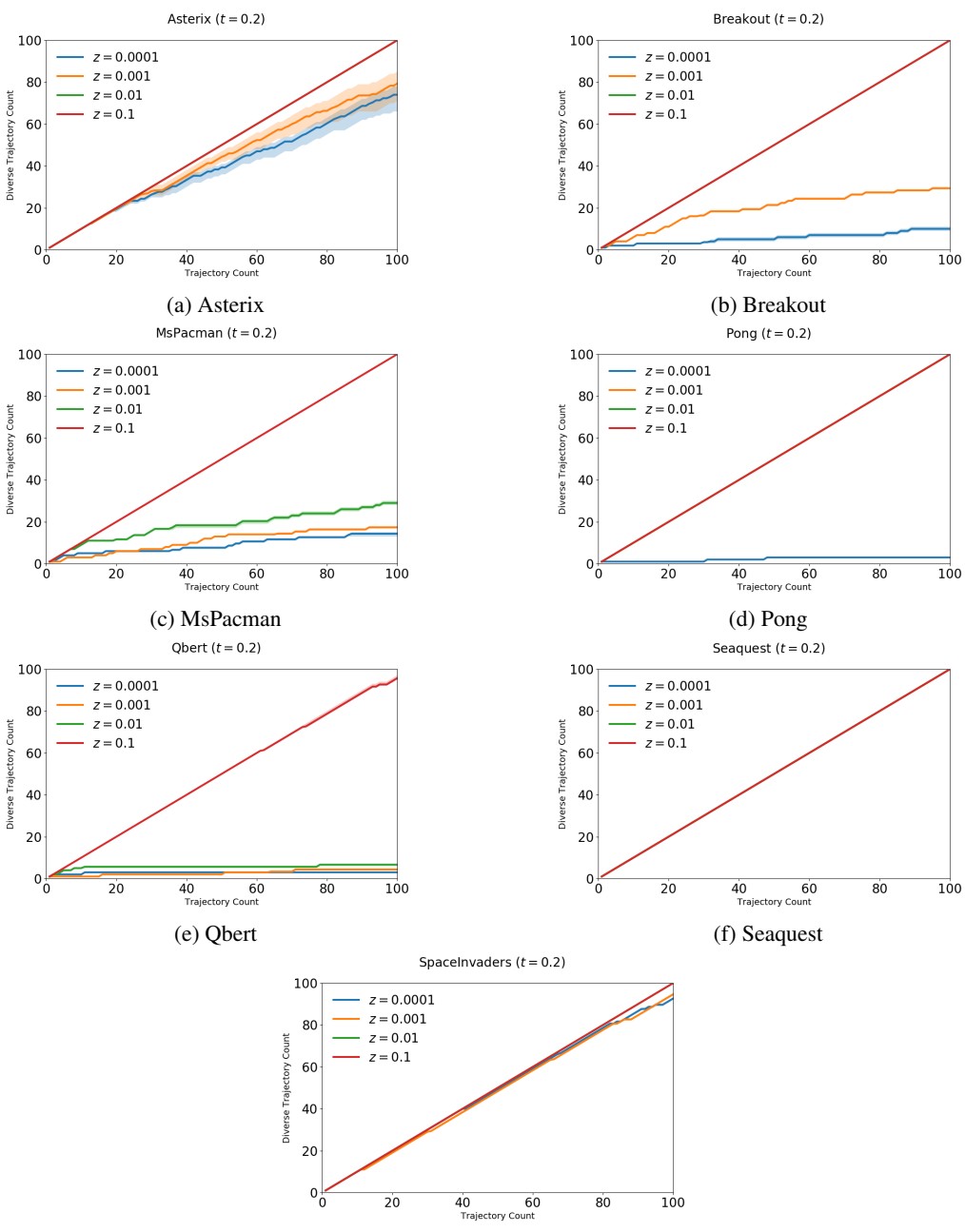

(a) Asterix

(b) Breakout

(c) MsPacman

(d) Pong

(e) Qbert

(f) Seaquest

(g) SpaceInvaders

Figure 15: Diversity Measurement of IQN Models in Seven Atari Games ($t = 0.2$)

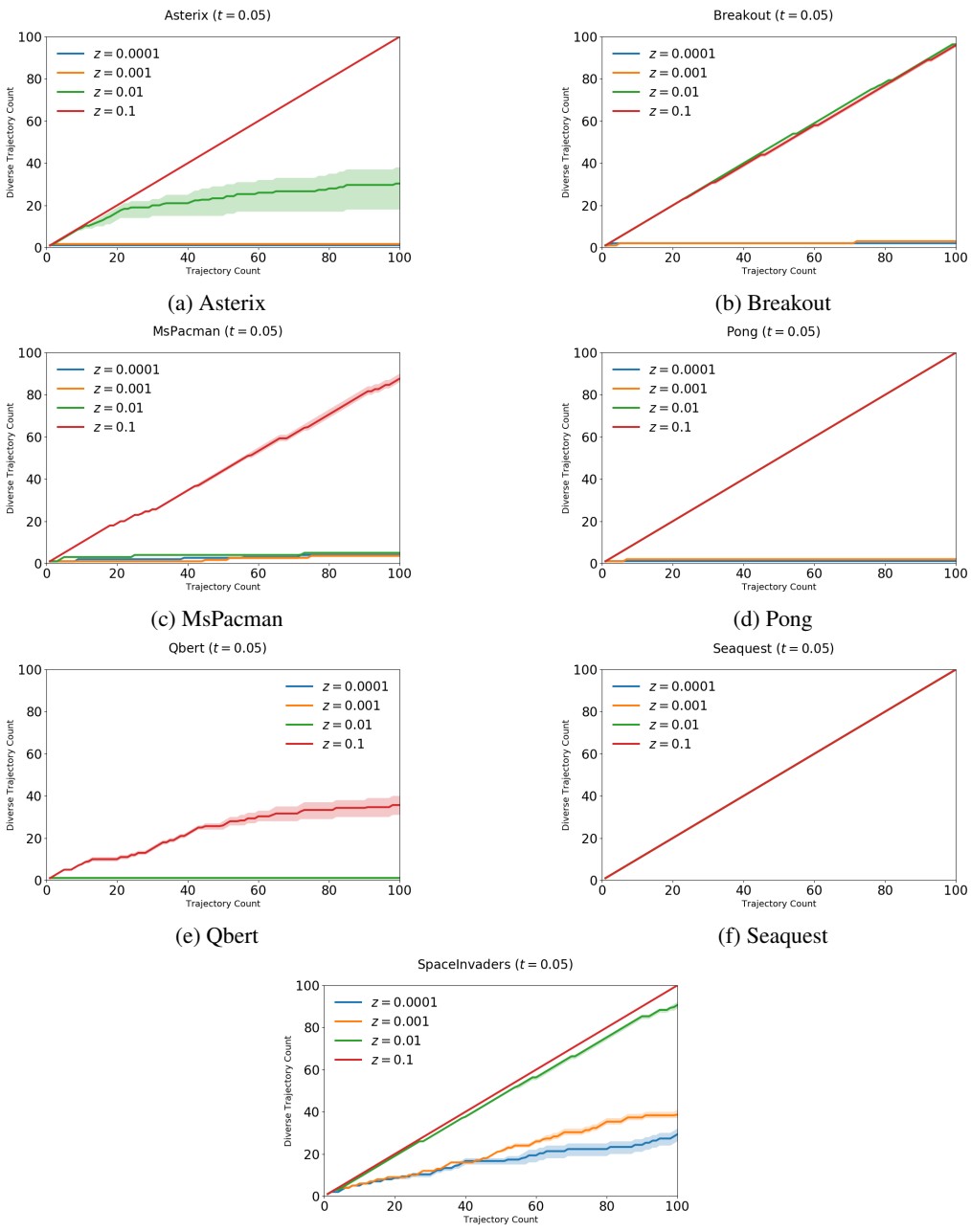

(a) Asterix

(b) Breakout

(c) MsPacman

(d) Pong

(e) Qbert

(f) Seaquest

(g) SpaceInvaders

Figure 16: Diversity Measurement of IQN Models in Seven Atari Games ($t = 0.05$)

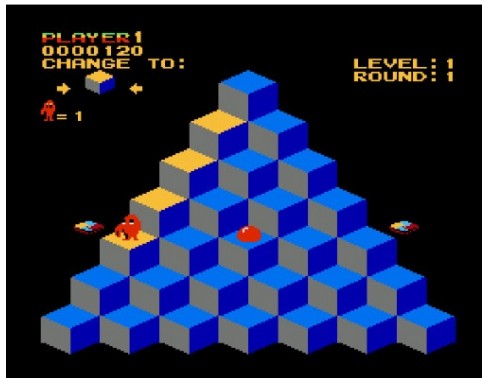 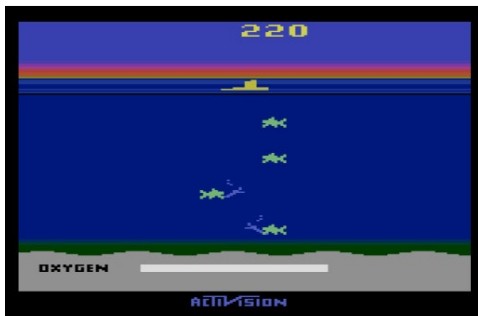

(a) Qbert: A 2D puzzle game where your goal is to change every cube in a pyramid to a target color. To do this, control the on-screen character, Q*bert, and make it jumps on top of the cube, avoiding obstacles and enemies.

(b) Seaquest: A 2D survival shooting game. The player sails a submarine to shoot at sharks and enemy submarines to rescue divers swimming in the water.

Figure 17: Game screens of Qbert and Seaquest.

### A.5.3 EXPECTED VERSION OF PLAYSTYLE JACCARD INDEX

$$
J_\Phi(M_A, M_B) = \frac{J_\Phi(M_A|M_B)}{2} + \frac{J_\Phi(M_B|M_A)}{2},
$$

$$
\text{where} \quad J_\Phi(M_X|M_Y) = \frac{|\bigcup_{o \in M_Y, \Phi(o) \in \Phi(M_X) \cap \Phi(M_Y)}[o]|}{|\bigcup_{o \in M_Y, \Phi(o) \in \Phi(M_X) \cup \Phi(M_Y)}[o]|}
$$

$$
= \frac{|\bigcup_{\phi \in \Phi} \bigcup_{o \in M_Y, \phi(o) \in \phi(M_X) \cap \phi(M_Y)}[o]|}{|\bigcup_{\phi \in \Phi} \bigcup_{o \in M_Y, \phi(o) \in \phi(M_X) \cup \phi(M_Y)}[o]|}
$$

(25)

### A.5.4 EXPECTED VERSION OF PLAYSTYLE SIMILARITY

$$
PS_\Phi^\cup(M_A, M_B) = \frac{PS_\Phi^\cup(M_A|M_B)}{2} + \frac{PS_\Phi^\cup(M_A|M_B)}{2},
$$

where $\quad PS_\Phi^\cup(M_X|M_Y) = \mathbb{E}_{o \sim M_Y, \bigcup_{\phi \in \Phi}\{\phi(o) \in \phi(M_X) \cup \phi(M_Y)\}}[P(D_\Phi^M(\pi_{M_X}(\phi(o)), \pi_{M_Y}(\phi(o))))]$

if $\pi_{M_X}(s)$ or $\pi_{M_X}(s)$ is undefined, $P(D_\Phi^M(\pi_{M_X}(s), \pi_{M_Y}(s))) = 0$

(26)

$$
D_\Phi^M(\pi_X, \pi_Y) = \frac{D(\pi_X, \pi_Y)}{\overline{D}_\Phi^{M,X}},
$$

where $\quad \overline{D}_\Phi^{M,X} = \mathbb{E}_{m \sim M-X}\Big[\frac{\mathbb{E}_{o \sim M_X, \bigcup_{\phi \in \Phi}\{\phi(o) \in \phi(M_X) \cap \phi(M_m)\}}[D(\pi_X(\phi(o)), \pi_m(\phi(o)))]}{2} +$

$\frac{\mathbb{E}_{\mathbb{E}_{o \sim M_m, \bigcup_{\phi \in \Phi}\{\phi(o) \in \phi(M_X) \cap \phi(M_m)\}}}[D(\pi_X(\phi(o)), \pi_m(\phi(o)))]}{2}\Big]$

(27)

