# OpenReview forum: "Perceptual Metrics for Video Game Playstyle Similarity and Diversity"
_ICLR.cc/2024/Conference — Submitted to ICLR 2024_

### Official Review · Reviewer_MAaz · 2023-10-23

**Soundness:** 2 fair
**Presentation:** 3 good
**Contribution:** 2 fair
**Rating:** 5
**Confidence:** 4

**Summary:**

The paper addresses the problem of defining a metric for comparing the similarity of playtraces from videogames (sequences of video frames and actions). The work improves the prior method called Playstyle Distance in three ways:
1) use of multiple encodings of states
2) use of an exponential kernel to scale the distance metric and treat it as a probability
3) use of the Jaccard index to improve assessment of overlap compared to the prior intersection

The evaluations assess each of these changes in isolation (to the degree possible), along with an overall assessment. Evaluations primarily used two datasets (one from humans, one from AI agents), with a third dataset used to evaluate the complete set of techniques. Results show the full metric improves over the prior Playstle Distance metric.

**Strengths:**

# originality
The paper focuses on a problem with limited prior work - developing ways to quantify similarity of playstyles. The originality of the work lies in the three improvements made to the Playstyle Distance metric. These are all reasonable incremental improvements.

# quality
Experiments vary single elements of the proposed changes to evaluate their efficacy; this provides some rigor in the evaluation of the overall method. The technical improvements are motivated by limitations of the Playstyle Distance and employ theoretically well-motivated adjustments.

# clarity
The paper articulates the insight behind each of the three core extensions to the previous algorithm and describes the changes clearly.

# significance
Playstyle metrics working from video (and action) data are valuable tools for a variety of applications including assessing "human-likeness", characterizing reinforcement learning algorithm behavior diversity, to stylometry work to model how humans play games. The paper targets a problem with wide reuse, while building on core components of a prior model.

**Weaknesses:**

# originality
The original contribution is limited to extending a previous model. This is in line with the intended contribution of the paper, and remains in line with the broader objective of behavioral playstyle metrics that require little manual tuning. Thus it is not a major weakness.

# quality
The experiments are only weakly supportive of individual components of the changed method. Section 5.3 (the full model) provides the clearest results showing superiority of the new technique over prior efforts. Results in section 5.1 and 5.2 seem to show improvements for only 1 domain (of 2 tested), where the two domains are quite similar in structure (though one uses humans and one AI agents). Below are more detailed questions on the evaluation.

# clarity
No substantial issues with clarity. Only minor clarification questions (below).

# significance
The primary weakness is the technique is only compared to a single prior effort. Thus there is no sense of how this technique compares to the state of the art in performance. This makes it hard to claim the technique has major significance, beyond the clear feature that it depends on very little manual tuning (of features, heuristics, thresholds, or other parameters).

**Questions:**

- What other baselines could be used for comparison in these games?
	- The empirical evaluations focus on ablations of the existing metric, but do not compare against any other metrics. This makes it difficult to assess how the current approach fares compared to alternative ways of conceptualizing playstyle measurement.
- Table 1:
	- Can multiple seeds or variations of the method be run to quantify uncertainty in the performance?
	- The text claims multiscale features ("mix") are superior. Looking at RGSK this result is not clear: the $2^20$ result for t=1 is quite close to mix for t=1. For TORCS the mix results are more clearly better (modulo the lack of any estimate of uncertainty).
	- Better results on 1 of 2 cases evaluated (TORCS) is not very strong evidence for superiority of the method. Perhaps this would be stronger with evidence from Atari as well?
- Figure 3:
	- As with table 1, is it possible to add uncertainty estimates to the results presented?
	- The results on probabilistic similarity again show clear results for TORCS, but more ambiguous outcomes for RGSK. Lacking any confidence intervals, the results in RGSK look very close to one another for all methods.
	- Is it possible the dataset is the root cause of these issues? TORCS is from AI agents, which may be easier to classify in style than humans (though it's not obvious a priori why this would be true).
- Figure 4: What are the shaded areas?

---

> ### Author Response · Authors · 2023-11-12
> **Questions**
>
> ## Questions
> * > What other baselines could be used for comparison in these games?
>     * Within the datasets provided by Playstyle Distance, it is challenging to identify more comparable baselines due to the lack of playstyle labels in the training datasets. Additionally, the dataset for TORCS and the Atari games contains only one playstyle, which makes contrastive learning impractical.
>     * > The empirical evaluations focus on ablations of the existing metric, but do not compare against any other metrics. This makes it difficult to assess how the current approach fares compared to alternative ways of conceptualizing playstyle measurement.
>         * Regarding the empirical evaluations focusing on ablations of the existing metric and not comparing against other metrics, we note that a possible unsupervised clustering method mentioned in Playstyle Distance shows less than 30% accuracy for a 5-class classification (compared to over 80% accuracy for Playstyle Distance). Playstyle Distance itself is claimed as the first metric of its kind for video game playstyles in its abstract, which is why we focused on improving this baseline.
>
>
> * > Table 1:
>     * > Can multiple seeds or variations of the method be run to quantify uncertainty in the performance? The text claims multiscale features ("mix") are superior. Looking at RGSK this result is not clear: the     result for t=1 is quite close to mix for t=1. For TORCS the mix results are more clearly better (modulo the lack of any estimate of uncertainty).
>         * Our experiments, including those using HSD, were conducted with three different HSD encoders, as provided in the official release of Playstyle Distance on PapersWithCode. Below is a detailed table with results from these three HSD encoders, including the corresponding standard deviations. If more detailed standard deviation data for each accuracy in the 100-round random subsampling is needed, we can conduct further experiments, though this will require additional time.
>
> | - | $1$ | $2^{20}, t=2$ | $2^{20}, t=1$ | $256^{res}, t=2$ | $256^{res}, t=1$ | $\textbf{mix}, t=2$ | $\textbf{mix}, t=1$ |
> | -------- | -------- |  -------- |  -------- |  -------- |  -------- |  -------- |  -------- |
> | TORCS | 35.16 | 66.52/69.40/76.12 (std=4.02) | 56.76/63.32/62.40 (std=2.90) | 6.36/3.72/3.72 (std=1.24) | 49.40/62.84/70.04 (std=8.55) | 73.28/71.40/79.64 (std=3.53) | 72.88/76.68/75.60 (std=1.60) |
> | RGSK | 80.14 | 79.92/78.71/76.54 (std=1.40) | 92.67/97.42/90.2 (std=3.00) | 5.58/5.08/6.21 (std=0.46) | 25.50/20.67/33.58 (std=5.33) | 91.75/83.33/90.46 (std=3.70) | 95.54/91.88/95.21 (std=1.65) |
>
> * > Better results on 1 of 2 cases evaluated (TORCS) is not very strong evidence for superiority of the method. Perhaps this would be stronger with evidence from Atari as well?
>     * The result shows that the baseline is sensitive to hyperparameters, and the hierarchical state version appears more stable in different scenarios.
> * > Figure 3:
>     * > As with table 1, is it possible to add uncertainty estimates to the results presented?
>         * The curves in Figure 3 include very narrow shaded regions representing the minimum and maximum values from the three HSD accuracies.
>     * > The results on probabilistic similarity again show clear results for TORCS, but more ambiguous outcomes for RGSK. Lacking any confidence intervals, the results in RGSK look very close to one another for all methods. Is it possible the dataset is the root cause of these issues? TORCS is from AI agents, which may be easier to classify in style than humans (though it's not obvious a priori why this would be true).
>         * Regarding the results on probabilistic similarity, it's true that the improvement on RGSK is minor. In TORCS, each playstyle is closely controlled, with slight variations in driving target speed and noise, necessitating a more precise playstyle metric.
>
> * > Figure 4: What are the shaded areas?
>     * The shaded areas represent the minimum and maximum values of the evaluation results from the three HSD encoders. These models were trained from different random seeds, as released in the official implementation of the Playstyle Distance paper on PapersWithCode.

---

> > ### Comment · Reviewer_MAaz · 2023-11-15
> >
> > Thank you for the responses! The clarifications on the shaded areas are helpful and it would help to include those in the text (please let me know if I missed where they were during my read).
> >
> > > Playstyle Distance itself is claimed as the first metric of its kind for video game playstyles in its abstract, which is why we focused on improving this baseline.
> >
> > I understand the novelty of the approach compared to prior methods, so it makes sense that no alternatives were compared.
> >
> > > Within the datasets provided by Playstyle Distance, it is challenging to identify more comparable baselines due to the lack of playstyle labels in the training datasets. Additionally, the dataset for TORCS and the Atari games contains only one playstyle, which makes contrastive learning impractical.
> >
> > Are there other game datasets that could be evaluated that were not evaluated in the original paper? Providing a clear demonstration of superior results would convince me of the strength of the method.
> >
> > From the results in the new table it looks like the 95% confidence intervals overlap for t=1 and mix. So it does seem like the improvements may be quite limited.

---

> > > ### Author Response · Authors · 2023-11-15
> > >
> > > Thank you for your response and for diligently participating in this discussion.
> > >
> > > We have identified 29 trajectories in the RGSK training dataset. Although this is a training dataset, it was solely used to train the discrete encoder and does not carry playstyle label information into the testing phase. This approach is also fair for compared baselines, as all use the same encoder. We will conduct additional experiments with this dataset.
> > >
> > > In addition to using the collected dataset from the baseline paper, we have also gathered new datasets to measure the diversity of DRL agents for our paper. In our settings (Section A.4), there are four different levels of stochasticity, and we have collected 100 trajectories for each level. We will use these trajectories to perform an additional playstyle identification task at the trajectory level.
> > >
> > > While these two extra experiments will take some time, they can be completed within the rebuttal period. If we finish the experiments, we will post a general comment to share the results with all reviewers.
> > >
> > > > From the results in the new table it looks like the 95% confidence intervals overlap for t=1 and mix. So it does seem like the improvements may be quite limited.
> > >
> > > The original accuracy in RGSK at $2^{20},t=1$ is already quite high, but it is sensitive to "t" without a hierarchical state. In the mix (hierarchical state) configuration, it is evident that neither $t=2$ nor $t=1$ significantly affects the prediction. This consistency provides clear evidence that our proposed method, which utilizes a hierarchical state, is superior and more general.

---

> > > > ### Comment · Reviewer_MAaz · 2023-11-22
> > > >
> > > > Thank you, I look forward to the new results.

---

> > > > > ### Author Response · Authors · 2023-11-23
> > > > >
> > > > > Thank you for your response.
> > > > >
> > > > > The new results have been posted in the comments.
> > > > > This includes further explanations of the continuous playstyle spectrum (see comment https://openreview.net/forum?id=hfAEEsIQ6D&noteId=dC8rGa4wZu) and more datasets for evaluation (see comment https://openreview.net/forum?id=hfAEEsIQ6D&noteId=Y8hOj9hZof).
> > > > > If you think that additional trajectory identification results in Atari games are necessary to differentiate between models and potentially increase the rating above 5, please let me know.
> > > > > In our recent explanations, we tested a single model with different levels of stochasticity.

---

### Official Review · Reviewer_EUGE · 2023-10-27

**Soundness:** 3 good
**Presentation:** 2 fair
**Contribution:** 2 fair
**Rating:** 3
**Confidence:** 4

**Summary:**

The authors propose extensions of a previous approach, "Playstyle Distance," which attempted to quantify different playstyles by DRL agents in applications such as video game play. The authors argue why different modifications to this approach are needed for improvement, such as multi-scale state encoding and incorporating other metrics. They provide experiments showing the improvements in playstyle accuracy classification attributable to these modifications.

**Strengths:**

- The authors do a good job of explaining previous work.
- The authors provided code to recreate the experiments.
- The derivations in the paper seem correct.

**Weaknesses:**

- The theoretical contribution is marginal, including the connection to video game play. The concept of "Playstyle Distance" itself does not seem to be specific to game play---it is a straightforward distance between policies using encoded state/action pairs [while "straightforward" is a feature (and not a bug) here, it raises the question as to why this paper needs to be framed in this language at all].
- As "Playstyle Distance" and related concepts are previous work, the additional theoretical contributions are minimal. Incorporating the Bhattacharyya distance is fine, but the connections made to human psychology are a bit flimsy. Additionally, the "Playstyle Similarity" metric is ad-hoc (which is isn't inherently bad); it's also not immediately clear to me (or at least it wasn't argued) if it's a metric, as the product of two metrics isn't a metric in general.
- Given the above points, the experimental section is not substantive enough. While the authors compare Playstyle Distance against their various improvements, they use classification accuracy to compare. However, taking a step back---this is now just a supervised learning problem. Why are any of these needed at all? If the _actual_ task is playstyle classification, then algorithms need to be compared against supervised learning approaches.
- While I appreciate the fact that code was provided, there weren't good instructions in the supplementary material zip file. Clicking the PapersWithCode link in the paper then brought me to a page that clearly has the author names and affiliations listed.

**Questions:**

- Is Playstyle Similarty a metric?

---

> ### Author Response · Authors · 2023-11-12
> **Concerns**
>
> ## Concerns
> * > The theoretical contribution is marginal, including the connection to video game play. The concept of "Playstyle Distance" itself does not seem to be specific to game play---it is a straightforward distance between policies using encoded state/action pairs (while "straightforward" is a feature (and not a bug) here, it raises the question as to why this paper needs to be framed in this language at all).
>     * Our distance metric still relies on the Wasserstein distance for measuring the effort required to change playstyles. The Bhattacharyya distance, as a variant, aligns with our proposed probability mapping function derived from the Weber-Fechner law (as detailed in Section A.1).
> * > As "Playstyle Distance" and related concepts are previous work, the additional theoretical contributions are minimal. Incorporating the Bhattacharyya distance is fine, but the connections made to human psychology are a bit flimsy. Additionally, the "Playstyle Similarity" metric is ad-hoc (which is isn't inherently bad); it's also not immediately clear to me (or at least it wasn't argued) if it's a metric, as the product of two metrics isn't a metric in general.
>     * Playstyle Similarity is indeed a metric, incorporating a conditional consideration of intersection over union. In Section 5.3, we explain that without the Jaccard index, the metric focuses only on intersection states, ignoring samples outside this intersection set. With only the Jaccard index, action distribution is not measured. Thus, it's not merely a product of two metrics, but a combination that leverages all observed state-action pairs, including:
>         * Playstyle Intersection Similarity: A metric of action distribution similarity in the intersection set.
>         * Playstyle Jaccard Index: A metric of all observed states.
>         * Playstyle Similarity: A comprehensive metric of all observed state-action pairs.
> * > Given the above points, the experimental section is not substantive enough. While the authors compare Playstyle Distance against their various improvements, they use classification accuracy to compare. However, taking a step back---this is now just a supervised learning problem. Why are any of these needed at all? If the actual task is playstyle classification, then algorithms need to be compared against supervised learning approaches.
>     * We use a classification task with a similarity metric to validate these metrics. In these cases, it can be assumed that all supervised learning approaches would exhibit random-level accuracy since there are no playstyle labels in the training dataset. Additionally, contrastive learning is impractical for the TORCS and Atari games datasets as they contain only one playstyle, making it unsuitable for such learning.
> * > While I appreciate the fact that code was provided, there weren't good instructions in the supplementary material zip file. Clicking the PapersWithCode link in the paper then brought me to a page that clearly has the author names and affiliations listed.
>     * If you encounter any issues with running our code in the Supplementary Material, we are available to assist. The PapersWithCode link in Section 4 directs to the official dataset and codes of "Playstyle Distance," which serves as our baseline. Our implementation is a clone from their Git repository, and we have ensured not to reveal our names and affiliations.

---

> > ### Comment · Reviewer_EUGE · 2023-11-15
> > **Response to rebuttal**
> >
> > Thank you, I appreciate your responses to my concerns.

---

> ### Author Response · Authors · 2023-11-12
> **Questions**
>
> ## Questions
> * Is Playstyle Similarty a metric?
>     * Yes, Playstyle Similarity is a metric comprising several aspects:
>         * Playstyle Intersection Similarity: Measures action distribution similarity in the intersection set.
>         * Playstyle Jaccard Index: Measures similarity across all observed states.
>         * Playstyle Similarity: Encompasses all observed state-action pairs for a comprehensive metric.

---

> > ### Comment · Reviewer_EUGE · 2023-11-15
> > **Response to Questions**
> >
> > Sorry, I meant the mathematical definition of metric (e.g. satisfying triangle inequality). Is that true?

---

> > > ### Author Response · Authors · 2023-11-15
> > >
> > > Thank you for your response and for bringing a mathematical perspective to this discussion.
> > >
> > > Regarding whether Playstyle Similarity is a metric in the mathematical sense:
> > > No, Playstyle Similarity is not a metric in the strict mathematical sense.
> > >
> > > Our metric is designed for measuring similarity and does not align with the mathematical definition of a distance metric.
> > > A strictly defined metric in mathematics must adhere to properties such as Positivity, Symmetry, and the Triangle Inequality.
> > > Commonly referred to similarity "metrics" are actually similarity measures and not metrics per mathematical standards. For instance, the Jaccard Index and cosine similarity, while often called "metrics," do not qualify as such in a strict mathematical context.
> > >
> > > If you think that this distinction is important or should be highlighted in the main paper, we can refer to our method as a 'measure' instead of a 'metric', or provide an explanation in Section 3. If you have any further concerns about this paper, please let me know.

---

### Official Review · Reviewer_crkG · 2023-10-30

**Soundness:** 3 good
**Presentation:** 3 good
**Contribution:** 2 fair
**Rating:** 6
**Confidence:** 2

**Summary:**

The authors propose a novel playstyle similarity metric, based on several extensions to the work of Lin et al. (2021). These modifications entail a multiscale metric, an exponential scaling, justified by psychophysics research and a probabilistic interpretation of the Jaccard index. The resulting variants are evaluated on two racing games and 7 Atari games, in a playstyle classification scenario.

**Strengths:**

The paper is well structured, explaining each modification in turn. Each modification is reasoned for and realized using proven concepts. The range of evaluation domains is diverse and all modifications are evaluated independently, showing that each is contributing to the whole.

**Weaknesses:**

Several statements are not clear to the reviewer or hard to parse:
- It is worth noting that Lin et al. (2021) were able to distinguish intersection states even with unprocessed screen pixels in Atari games. - Seems to state "In two different datasets, Lin et al. (2021) did find pixel-identical screens"?
-  In a different scenario, when treating each state as equivalent, we can invariably pinpoint an intersection state. - Relates to identical, not just equivalent?
- Unclear what exactly is new in Sec. 3.1 and what was introduced as part of HDS (Lin et al. (2021)).
- While distance is a common metric for determining similarity, a larger distance value conveys primarily that two entities are different, without giving much insight into the degree of their similarity - Why doesn't a larger distance relates to a smaller degree of similarity?
- Drawing from the concept of similarity, we can infer that a smaller distance provides more definitive information about the similarity - Needs to be explained.
- General wording: "intersecting states/samples" suggest a partial equivalence of a single state/sample. E.g. "intersecting set of states/samples" could be more appropriate. This is especially relevant, because the HSD model can be used to defined a intersection over (single) states (based on the hierarchy), and therefore the concepts are not clearly distinguishable.
- Sec 4.2.: "Space size" seems not to be introduced - probably number of discrete states?
- To evaluate the efficacy of the proposed multiscale state space and to compare it fairly with Playstyle Distance, we primarily focus on the TORCS and RGSK platforms - Needs to be explained.
- In this section, we perform a comprehensive evaluation of various metrics, including leveraging full data with union operations. - Seems to state, that state samples from all games are used as a single set?

Besides these clarity issues, the experimental Section could use some improvements:
- This is based on the assumption that variations in game content can be interpreted as different states - This statement was not empirically evaluated?
- The "game-merging" study in Sec 5.3. is an interesting piece of additional information, but the per-game results are potentially more relevant. The results should be added to the main paper and in case of space constraints, one may think about rank metrics or a table with just the most interesting dataset sizes.
- Why was Playstyle Similarity not evaluated in Sec. 5.2?
- Most importantly, additional comparison baselines from related work should be added. E.g. clustering or supervised methods should be applicable.
- The evaluation is only performed via classification, but a similarity metric should preserve a distance relation beyond "Top-1". Therefore, a ranking/continuous study should be performed. E.g. by creating or ordering existing playstyles along a continuous spectrum (like passive/aggressive driving) and evaluating the correlation.
- Sec 5.2 a, b has nearly indistinguishable intervals. A tabular presentation or non-linear graph scaling should be used to improve clarity.

Overall, the evaluation is still strong, but additional baselines and per-game results would greatly contribute to its value. The contribution was only deemed fair, mostly because the work does only modify an existing idea and the topic is quite niche.

**Questions:**

See confirmative questions above.

---

> ### Author Response · Authors · 2023-11-12
> **Concerns Part1**
>
> ## Concerns
> * > It is worth noting that Lin et al. (2021) were able to distinguish intersection states even with unprocessed screen pixels in Atari games. - Seems to state "In two different datasets, Lin et al. (2021) did find pixel-identical screens"?
>     * In Atari games, pixel-identical consecutive screens can exist in two game trajectories due to the simplicity of the game environment and low randomness. In common DRL applications, we often find almost no repeated visited states due to high-dimensional observations.
> * > In a different scenario, when treating each state as equivalent, we can invariably pinpoint an intersection state. - Relates to identical, not just equivalent?
>     * When discretized states are considered, they can be treated as identical. However, when a state has a corresponding action, we do not eliminate the same state-action pair for constructing the sampling action distribution.
> * > Unclear what exactly is new in Sec. 3.1 and what was introduced as part of HDS (Lin et al. (2021)).
>     * Lin et al. (2021) used a single discrete state mapping and a sample count threshold to discard states with unstable action distributions. They did not use the hierarchical design in HSD for Playstyle Distance as it prefers stable action distributions and not very large discrete state spaces. Our new method in Section 3.1 enables the simultaneous use of different discrete state mappings.
> * > While distance is a common metric for determining similarity, a larger distance value conveys primarily that two entities are different, without giving much insight into the degree of their similarity - Why doesn't a larger distance relates to a smaller degree of similarity? Drawing from the concept of similarity, we can infer that a smaller distance provides more definitive information about the similarity - Needs to be explained.
>     * As the distance increases, the number of potential candidates with the same distance also increases significantly. In a 2D space, these candidates form a circle around a reference point; as the distance increases, the perimeter of the circle increases. Therefore, we focus more on the similar parts for a more definitive prediction.
>
> * > General wording: "intersecting states/samples" suggest a partial equivalence of a single state/sample. E.g. "intersecting set of states/samples" could be more appropriate. This is especially relevant, because the HSD model can be used to defined a intersection over (single) states (based on the hierarchy), and therefore the concepts are not clearly distinguishable.
>     * We can edit this part for a more precise description as long as the page limit is not exceeded in the camera reday version.
> * > Sec 4.2.: "Space size" seems not to be introduced - probably number of discrete states?
>     * The term "Space size" refers to the maximum count of different states in the discrete representation. We will use the term "state space" for more precise descriptions.
> * > To evaluate the efficacy of the proposed multiscale state space and to compare it fairly with Playstyle Distance, we primarily focus on the TORCS and RGSK platforms - Needs to be explained.
>     * We validate Playstyle Distance with solely multiscale state space on TORCS and RGSK to make sure multiscale state space is effective. For Atari games, single state space version already has a high accuracy with 512 sample size (over 90% accuracy). With the page limit, we move this part with curves to appendix (Figure 11 and 12).
> * > In this section, we perform a comprehensive evaluation of various metrics, including leveraging full data with union operations. - Seems to state, that state samples from all games are used as a single set?
>     * In Sections 5.1 and 5.2, our focus was on the observed state in the intersecting set of two datasets. In Section 5.3, we expanded this to include all observed states. The introduction of the Jaccard index allows us to define action distribution similarity even outside the intersecting set of samples.

---

> ### Author Response · Authors · 2023-11-12
> **Concerns Part2**
>
> * > This is based on the assumption that variations in game content can be interpreted as different states - This statement was not empirically evaluated?
>     * This assumption was validated using the Atari Console, employing a discrete encoder from other games.
> * > The "game-merging" study in Sec 5.3. is an interesting piece of additional information, but the per-game results are potentially more relevant. The results should be added to the main paper and in case of space constraints, one may think about rank metrics or a table with just the most interesting dataset sizes.
>     * The per-game results of Atari games in Section 5.3 have been moved to Section A.3. We can add a table in the appendix to report accuracy values. If necessary, we can consider removing or moving parts of the main paper to accommodate this table.
> * > Why was Playstyle Similarity not evaluated in Sec. 5.2?
>     * In Section 5.2, our focus remained on states within the intersection. The concept of Playstyle Similarity, which includes the Jaccard index, is discussed in Section 5.3, as it encompasses all observed samples.
> * > Most importantly, additional comparison baselines from related work should be added. E.g. clustering or supervised methods should be applicable.
>     * Common methods like the Fréchet Inception Distance (FID) use latent feature distribution distances, which have been deemed impractical for classification in TORCS (as shown in Playstyle Distance). Supervised or contrastive learning baselines are challenging due to the absence of playstyle labels in training datasets or the presence of only one playstyle in the dataset.
> * > The evaluation is only performed via classification, but a similarity metric should preserve a distance relation beyond "Top-1". Therefore, a ranking/continuous study should be performed. E.g. by creating or ordering existing playstyles along a continuous spectrum (like passive/aggressive driving) and evaluating the correlation.
>     * We plan to use the TORCS dataset for an experiment evaluating continuous spectrum playstyles (5 target speeds from 60 to 80, and 5 levels of noise). This experiment will be conducted during the rebuttal period.
> * > Sec 5.2 a, b has nearly indistinguishable intervals. A tabular presentation or non-linear graph scaling should be used to improve clarity.
>     * We will include tables in the appendix to provide a clearer presentation of the results from Figure 3(a)(b).

---

> > ### Comment · Reviewer_crkG · 2023-11-16
> >
> > Thanks for the detailed answers. They clarify several of my questions and i also acknowledge the additional results. They are also not directly clear to me, but this issue was already raise by another reviewer and does not need any further discussion from my side. Assuming the clarifications will be introduced into a potential CR version, i will not argue against an accept (while already being slightly in favor anyhow).

---

> > > ### Author Response · Authors · 2023-11-16
> > >
> > > Thank you for your response and for acknowledging our contribution.
> > > We are conducting additional experiments to provide more convincing results.

---

### Official Review · Reviewer_6U4t · 2023-10-31

**Soundness:** 3 good
**Presentation:** 2 fair
**Contribution:** 1 poor
**Rating:** 3
**Confidence:** 3

**Summary:**

The paper addresses the problem of discerning similarities among datasets containing state/action pairs derived from video games. This aids in pinpointing distinct playstyles and understanding the diversity of human behaviors within these games. The work enhances existing methods, particularly the Playstyle Distance technique, by modifying this approach. This traditional method involves an initial discretization of the state space, followed by a comparison of action distributions based on these discrete states using a wasserstein distance. The authors propose three advancements. Firstly, they introduce multiscale states, a refined discretization technique that uses a combination of mappings rather than just one. Secondly, they use the Bhattacharyya distance as an alternative to the Wasserstein distance, arguing that the former aligns more with human perception. Lastly, the authors use the Jaccard index to weight the distance when comparing conditional action distributions across intersecting states in both reference and analyzed datasets, ensuring a more accurate assessment of the intersection's significance. The researchers conducted experiments on three distinct games: two racing games and a collection of Atari games. When compared with various versions of the Playstyle Distance, their proposed methodology appears more proficient. It more effectively captures similarities, leading to a more accurate identification of players.

**Strengths:**

The proposed variations over existing approaches sound interesting and grounded on human cognition. By building upon the foundations laid out by prior research, this paper takes strides in refining and enhancing what has been previously suggested in the domain. It is a simple approach that performs well on the described datasets and that extends the possible playstyle identification set of methods.

**Weaknesses:**

I find it challenging to grasp the exact task the authors aim to address, even though I recognize it's based on prior work. The paper's objective seems to be playstyle identification based on a few gameplay samples by comparing them with reference datasets,, but the experiments are more about player identification. While understanding the need for reference datasets for different playstyles, the creation and existence of these references remain ambiguous. A discussion on this would be beneficial.
Moreover, the assumption is that the more the distributions of actions are different, the further the playstyles are. But is it really waht defines playstyle? What about identifying playstyles that are only different in very few states (for instance, two chess players that are using two different openings). Since the definition of palystyle provided here is not grounded on any concrete application, it is difficult to understand the relevance of the work.

Second, the approach hinges on discretizing the state space. While states are assumed to be continuous and actions discrete, how would this apply when actions are also continuous, as seen in many games? The method's efficacy seems tied to the state discretization's capability to reflect genuine state distances. This might work for pixel-based games, but what about more structured observations, like chess? The multiscale approach's specifics, including the number of mapping functions and their selection, are unclear, yet these details likely influence the outcome significantly.

Last, the approach is an unsupervised method and is evaluated on very few use-cases. It makes it difficult to understand if the proposed approach is good 'in general' or if the choices made by the authors have been 'over-fitted' on the three single use cases they propose. The paper lacks a real dataset captured from real video games, with complex states, actions, and many more players than what is in the article.

If the contribution sounds right and improves over existing publications, the validation does not allow us to conclude if the approach is really good for identifying playstyles or not. The article lacks a clear definition of what a playstyle is, and why identifying playstyles is interesting. While the subject may appeal to a niche audience, particularly those in the video game research community  (e.g Cog conference), it might not meet the broader criteria for acceptance at ICLR

**Questions:**

* What are the real use-cases that the identification of playstyle is targeting? Why is it an unsupervised problem and not a supervised one ? How the playstyle references dataset are built and is it realistic?
* What is the effect of the state discretization technique that is used ? How do you tune the multiscale approach ? Since you are only using few datasets, the way you tune it may be overfitting the datasets, isn't it ?
* How do you deal with continuous actions since your work focuses on discrete action spaces?

**Details Of Ethics Concerns:**

No concerns

---

> ### Author Response · Authors · 2023-11-12
> **Concerns**
>
> ## Concerns
> ### In summary
> * Our distance is still based on Wasserstein distance for measuring the effort between changing playstyle. Bhattacharyya distance is a variant that meets our proposed probability mapping function derived form Weber-Fechner law (Section A.1).
> ### In weakness
> * The objective of this paper is to define a similarity metric. The experiments for validating this metric involve identification/classification tasks and the application of diversity measurement in DRL agents. The reference datasets used in the experiments serve as examples of targeted playstyles. For instance, to measure an unknown playstyle, we can compare it against known playstyles and identify the most similar one. This scenario is akin to a zero-shot classification task, where no classification task occurs during the training of discrete representations.
> * For new games without existing reference sets, it's possible to first collect some trajectories and label them manually (giving meaning to the playstyle) or use a similarity threshold to create reference datasets.
> * Regarding concerns about very few intersection states (as in chess), this highlights why discrete state representation is important for determining comparable cases. If the playstyle (decision-making style) is indeed different, it should show action distribution differences in those intersection states (like opening or initial board). If the playstyle is very similar yet leads to different states, the result is either influenced by environmental randomness or decisions made by other players, which are not under the control of the player we are measuring. Otherwise, the same action in the same state should lead to the same next state, making it unlikely to have very few intersection states.
> * Regarding the concern about continuous action spaces, TORCS, as described in Section 4.1, is a game with a continuous action space.
> * The multiscale approach is rooted in human cognition and involves several levels of granularity. In Table 1 and Section A.2, we demonstrate that each state mapping function contributes some information to playstyle measurement (accuracy > $\frac{1}{\text{number of playstyles}}$). The combined version further improves accuracy (exceeding the use of just one state mapping). As this research is not focused on training discrete states, we directly utilize the hierarchical states from $\textit{Playstyle Distance}$ and also test with downsampling discretization. Our experiments show that more state mappings lead to more accurate predictions. These mappings can be obtained by training more discrete encoders or leveraging HSD (simultaneous training of several discrete encoders with customized granularity).
> * Regarding the concern about generality, our metric is built on general decision-making and tested on rule-based agents, human players, and DRL agents. Our experiments are performed on video games, and we claim applicability to video games due to the straightforward scenario of demonstrating multiple playstyles. Should open benchmarks for decision-making styles become available, we believe our approach would be suitable for claiming broader generality. These decision-making problems could include robot control, natural language processing, and others, as described in the introduction, where DRL is a solution for general decision-making.

---

> ### Author Response · Authors · 2023-11-12
> **Questions**
>
> ## Questions
> * > What are the real use-cases that the identification of playstyle is targeting?
>     * Our metrics are designed to concretely define the similarity and diversity of decision-making styles (playstyles). They are useful in validating the diversity of AI policies, assessing whether imitation learning fits a given playing dataset, and for human behavior modeling. For instance, they can detect the similarity of a Go player's playstyle to that of the superhuman AlphaZero AI to prevent cheating. Playstyle metrics are also useful in recommendation systems for identifying potential customers with similar or new playstyles.
> * > Why is it an unsupervised problem and not a supervised one ?
>     * The metric and discrete representation are unsupervised, meaning no playstyle labels are used in training. This approach is akin to zero-shot classification, with no learning how to classify during the discrete representation learning phase. States are used for identifying comparable states for action distribution comparison, which implies playstyle information.
>     * According to Appendix D of the HSD paper (Playstyle Distance), it is impractical to perform supervised learning on training the discrete representation or playstyle classification, as either the datasets are sampled from different sources (TORCS and RGSK) or there is only one playstyle (Atari games).
>     * Besides, as described in Section 2.1, playstyle classification can be trained by supervised learning, but it relies on predefined labels and cannot handle playstyles not in the training data (the datasets we used cannot be solved by supervised learning).
> * > How the playstyle references dataset are built and is it realistic?
>     * The datasets used in playstyle analysis consist of gameplay collected by rule-based agents, human players, and DRL agents. For performing few-sample analysis, we use random subsampling without replacement, as shown in Figure 2(d).
>     * As for the realism of the games used in our experiments, TORCS and Atari games are common in DRL research. RGSK is a racing game available on the Unity Asset Store (https://assetstore.unity.com/packages/templates/systems/racing-game-starter-kit-22615), designed for human players.
> * > What is the effect of the state discretization technique that is used?
>     * The state discretization technique identifies comparable states for comparing action distributions. Without state discretization, it is challenging to measure the action distribution distance directly from playing datasets (observation-action pairs).
>     * In Section A.2, our experiments show that the state discretization technique we used (HSD) performs better than downsampling. The learned discrete representation focuses on features more related to gameplay rather than arbitrary features.
> * > How do you tune the multiscale approach ? Since you are only using few datasets, the way you tune it may be overfitting the datasets, isn't it?
>     * We do not tune the multiscale approach, as it does not require any hyperparameters for our proposed metric. The elements that can be tuned are the discrete state mappings. However, the scope of this paper is not about training new discrete representations; therefore, we directly use the trained models released by the baseline method (Playstyle Distance).
> * > How do you deal with continuous actions since your work focuses on discrete action spaces?
>     * Our metrics can handle both discrete and continuous action spaces, with the difference being the adoption of different types of probability distribution distance metrics. Specifically, TORCS has a continuous action space, while RGSK and Atari games have discrete action spaces, as described in Section 4.1.

---

### Author Response · Authors · 2023-11-12
**To all reviewers**

Dear Reviewers,

Thank you for dedicating your time and effort to review our manuscript.

We will address the general concerns here and respond to all specific concerns and questions directed at each reviewer with another official comment to each official review.

We noticed that all reviewers brought up a question regarding why this task cannot be solved with supervised learning or more potential baselines. A key point that may have been unclear or missed in our presentation is the absence of playstyle labels in the training datasets. Moreover, the TORCS training dataset is hard to split into more than one playstyle, and the playstyles in Atari games are essentially the same (as indicated in the appendix of the baseline paper - Playstyle Distance, each Atari game training dataset is sampled from a single model). Hence, the playstyle identification/classification tasks in our experiments can be considered as zero-shot classification tasks, with no playstyle label information in the training set. Within this setting, it is impractical to perform supervised learning, or the result would be trivial, akin to a random model due to the absence of training labels. Contrastive learning is also impractical since it relies on identifying self-class and other classes (at least two classes in a dataset), but only one class exists in the training dataset.

We can reference two possible baseline papers: Playstyle Distance metric (Lin et al., 2021) and Behavioral Stylometry (McIlroy-Young et al., 2021). Both papers claimed no comparable baseline for this type of problem, with Playstyle Distance further stating in their abstract that it is the first metric of this kind. We did not find many methods for solving this type of problem; the most relevant are Playstyle Distance and Behavioral Stylometry, and the latter requires some playstyle information during model training. This is why our method does not include more baselines besides Playstyle Distance. We could add a supervised or contrastive learning baseline, but we think this would not provide much insight since they would essentially be random models.

Another general concern is about what playstyle is and whether it is valuable to the ICLR community.
Actually, playstyle refers to decision-making style. However, its value to a domain may depend on the use cases. For decision problems with only one optimal solution, style may not be necessary. The most pursuing decision style domain in our knowledge is gaming, as it is well-known for entertainment. Our playstyle (decision-making style) metric can be extended to general decision-making, which only uses observation-action pairs. We believe that as more decision-making problems solved by DRL, such as robot control, NLP, among others, become more popular and focus on styles after achieving decent performance (e.g., human-level), this will underline why it is an important topic for the ICLR community.
Since our experiments only use video games, we only claim applicability to video games, but actually, it is a general decision-making style metric.

Additionally, Reviewer crkG suggests performing a continuous spectrum experiment beyond “Top-1” for a metric. We think this is an excellent suggestion to validate the metric and we plan to run this experiment in the coming days. In TORCS, there are 5 speed styles and 5 action noise styles in the testing datasets; we believe we can build a 5x5 similarity matrix to analyze the metric.

---

> ### Author Response · Authors · 2023-11-13
>
> The continuous playstyle spectrum experiments have been completed. These results will be added to a new section of the appendix if any reviewer agrees that it can help increase the quality of this paper. More detailed experiments about the continuous playstyle spectrum may be conducted for the final version (camera-ready version).
>
> # Experiment Settings
> **Goal**: To analyze whether the metric values (similarity) consistently change as the continuous playstyle changes.
>
> **Game**: TORCS
> **Discrete Representation**: The first models of HSD. (There are a total of 3 models.)
> **Dataset**:
> * Conducted 100 rounds of random subsampling, with each sampled dataset being disjoint and consisting of 512 observation-action pairs.
> * According to the baseline paper (Playstyle Distance), there are 5 levels of target speeds (60, 65, 70, 75, and 80) and 5 levels of action noises applied to the 2-dimensional actions: ((0.01,0.005), (0.02,0.01), (0.03,0.015), (0.04,0.02), and (0.05,0.025)).
>
> **Compared Metrics**:
> * Playstyle Distance (discrete state space: 2^20, the baseline method)
> * Playstyle Distance (discrete state space: mix)
> * Playstyle Intersection Similarity (discrete state space: mix)
> * Playstyle Jaccard Index (discrete state space: mix)
> * Playstyle Similarity (discrete state space: mix)
>
> If the similarity consistently decreases as the continuous playstyle changes significantly, we mark a (C) for counting the consistency of a row or column of the playstyle. The higher the consistency count, the better. We also record the standard deviation of metric values in 100 rounds of random subsampling, denoted as (std) in the table.
>
> If the similarity is strictly decreased as the continuous playstyle changed more, we mark a (C) for counting the consistency of a row or column of the playstyle.
> The higher the consistency count, the better. We also record the standard deviation of metric values in 100 rounds of random subsampling, denoted as (std) in the table.

---

> > ### Author Response · Authors · 2023-11-13
> >
> > # Test1 (from corner)
> > Compute similarity between the target playstyle, Speed60N0 (driving with target speed 60 and (0.01,0.005) action noise), and all playstyles (a total of 25 playstyles).
> >
> > * Playstyle Distance (2^20): consistent count = 3
> >
> > | Speed60N0 | Speed60 (C) | Speed65 | Speed70 | Speed75 | Speed80 |
> > | -------- | -------- | -------- | -------- | -------- | -------- |
> > | N0 (C) | -0.0044 (0.0012) | -0.0048 (0.0010) | -0.0055 (0.0010) | -0.0069 (0.0011) | -0.0100 (0.0014) |
> > | N1 (C) | -0.0054 (0.0011) | -0.0064 (0.0015) | -0.0069 (0.0016) | -0.0086 (0.0013) | -0.0108 (0.0016) |
> > | N2 | -0.0064 (0.0010) | -0.0058 (0.0010) | -0.0068 (0.0013) | -0.0086 (0.0010) | -0.0097 (0.0015) |
> > | N3 | -0.0073 (0.0011) | -0.0078 (0.0013) | -0.0071 (0.0010) | -0.0100 (0.0017) | -0.0118 (0.0016) |
> > | N4 | -0.0092 (0.0013) | -0.0083 (0.0011) | -0.0089 (0.0013) | -0.0110 (0.0015) | -0.0132 (0.0017) |
> >
> > * Playstyle Distance (mix): consistent count = 5
> >
> > | Speed60N0 | Speed60 (C) | Speed65 | Speed70 | Speed75 (C) | Speed80 |
> > | -------- | -------- | -------- | -------- | -------- | -------- |
> > | N0 (C) | -0.0018 (0.0005) | -0.0022 (0.0005) | -0.0028 (0.0006) | -0.0037 (0.0006) | -0.0060 (0.0007) |
> > | N1 (C) | -0.0024 (0.0005) | -0.0028 (0.0007) | -0.0036 (0.0008) | -0.0043 (0.0007) | -0.0058 (0.0008) |
> > | N2 | -0.0028 (0.0005) | -0.0027 (0.0006) | -0.0031 (0.0006) | -0.0045 (0.0007) | -0.0056 (0.0008) |
> > | N3 (C) | -0.0030 (0.0005) | -0.0033 (0.0007) | -0.0035 (0.0006) | -0.0049 (0.0007) | -0.0060 (0.0009) |
> > | N4 | -0.0038 (0.0006) | -0.0037 (0.0006) | -0.0040 (0.0006) | -0.0055 (0.0008) | -0.0071 (0.0011) |
> >
> > * Playstyle Intersection Similarity (mix): consistent count = 8
> >
> > | Speed60N0 | Speed60 (C) | Speed65 (C) | Speed70 (C) | Speed75 (C) | Speed80 (C) |
> > | -------- | -------- | -------- | -------- | -------- | -------- |
> > | N0 (C) | 0.8014 (0.0206) | 0.7502 (0.0187) | 0.7170 (0.0233) | 0.6538 (0.0231) | 0.5829 (0.0246) |
> > | N1 (C) | 0.6927 (0.0234) | 0.6865 (0.0240) | 0.6646 (0.0218) | 0.6254 (0.0265) | 0.5508 (0.0250) |
> > | N2 | 0.6260 (0.0266) | 0.6499 (0.0257) | 0.6354 (0.0299) | 0.5709 (0.0254) | 0.5450 (0.0284) |
> > | N3 | 0.5813 (0.0282) | 0.5857 (0.0250) | 0.5721 (0.0302) | 0.5507 (0.0276) | 0.4825 (0.0321) |
> > | N4 (C) | 0.5420 (0.0268) | 0.5390 (0.0298) | 0.5322 (0.0310) | 0.4708 (0.0288) | 0.4544 (0.0328) |
> >
> > * Playstyle Jaccard Index (mix): consistent count = 5
> >
> > | Speed60N0 | Speed60 (C) | Speed65 | Speed70 (C) | Speed75 (C) | Speed80 (C) |
> > | -------- | -------- | -------- | -------- | -------- | -------- |
> > | N0 (C) | 0.0938 (0.0059) | 0.0863 (0.0042) | 0.0841 (0.0044) | 0.0840 (0.0048) | 0.0816 (0.0042) |
> > | N1 | 0.0845 (0.0046) | 0.0853 (0.0045) | 0.0833 (0.0040) | 0.0821 (0.0043) | 0.0815 (0.0040) |
> > | N2 | 0.0827 (0.0045) | 0.0816 (0.0037) | 0.0830 (0.0047) | 0.0812 (0.0043) | 0.0799 (0.0042) |
> > | N3 | 0.0821 (0.0040) | 0.0820 (0.0048) | 0.0790 (0.0042) | 0.0806 (0.0040) | 0.0773 (0.0049) |
> > | N4 | 0.0780 (0.0046) | 0.0762 (0.0045) | 0.0757 (0.0038) | 0.0750 (0.0042) | 0.0760 (0.0041) |
> >
> > * Playstyle Similarity (mix): consistent count = 9
> >
> > | Speed60N0 | Speed60 (C) | Speed65 (C) | Speed70 (C) | Speed75 (C) | Speed80 (C) |
> > | -------- | -------- | -------- | -------- | -------- | -------- |
> > | N0 (C) | 0.0753 (0.0046) | 0.0650 (0.0035) | 0.0608 (0.0034) | 0.0555 (0.0026) | 0.0472 (0.0035) |
> > | N1 (C) | 0.0590 (0.0035) | 0.0579 (0.0033) | 0.0549 (0.0034) | 0.0513 (0.0035) | 0.0446 (0.0031) |
> > | N2 | 0.0519 (0.0036) | 0.0533 (0.0033) | 0.0525 (0.0033) | 0.0458 (0.0032) | 0.0440 (0.0034) |
> > | N3 (C) | 0.0482 (0.0033) | 0.0473 (0.0034) | 0.0448 (0.0031) | 0.0435 (0.0032) | 0.0382 (0.0030) |
> > | N4 (C) | 0.0421 (0.0034) | 0.0413 (0.0031) | 0.0402 (0.0035) | 0.0349 (0.0030) | 0.0337 (0.0032) |

---

> > ### Author Response · Authors · 2023-11-15
> >
> > Let me offer a more detailed explanation of the consistency count using algebraic notations.
> > The concept of consistency in this context is based on the sorting of playstyles using the playstyle metric. The metric values should yield not only correct predictions for the Top-1 similarity but also a correctly sorted sequence according to the metric values.
> >
> > In the continuous playstyle spectrum experiments, we assume that the playstyle can change continuously. To explain more concretely, consider five playstyles A, B, C, D, E, and a target playstyle A' that belongs to playstyle A. We hypothesize that Similarity(A',A) > Similarity(A',B) > Similarity(A',C) > Similarity(A',D) > Similarity(A',E). Therefore, for a similarity metric $M$, we should observe $M(A',A)$ > $M(A',B)$ > $M(A',C)$ > $M(A',D)$ > $M(A',E)$. If this is not the case, then the similarity metric fails the consistency test.
> >
> > In "Test2 (from center)", we consider the same five playstyles and a target playstyle C' belonging to playstyle C. Since there are two directions of change relative to C', the following two sequences should be observed for consistency:
> > 1. Similarity(C',C) > Similarity(C',B) > Similarity(C',A)
> > 2. Similarity(C',C) > Similarity(C',D) > Similarity(C',E)
> >
> > Thus, the similarity metric must follow the corresponding relation to satisfy the consistency:
> > 1. $M(C',C)$ > $M(C',B)$ > $M(C',A)$
> > 2. $M(C',C)$ > $M(C',D)$ > $M(C',E)$
> >
> > A higher consistency count implies that the playstyle metric maintains better value consistency across different playstyles.

---

> ### Author Response · Authors · 2023-11-13
>
> # Test2 (from center)
> Compute similarity between the target playstyle, Speed70N2 (driving with target speed 70 and (0.03,0.015) action noise), and all playstyles (a total of 25 playstyles).
>
> * Playstyle Distance (2^20): consistent count = 2
>
> | Speed70N2 | Speed60 | Speed65 | Speed70 | Speed75 | Speed80 |
> | -------- | -------- | -------- | -------- | -------- | -------- |
> | N0 | -0.0068 (0.0013) | -0.0068 (0.0012) | -0.0065 (0.0011) | -0.0068 (0.0012) | -0.0097 (0.0017) |
> | N1 | -0.0073 (0.0011) | -0.0069 (0.0013) | -0.0073 (0.0014) | -0.0084 (0.0014) | -0.0095 (0.0014) |
> | N2 (C) | -0.0079 (0.0010) | -0.0075 (0.0015) | -0.0069 (0.0013) | -0.0087 (0.0014) | -0.0095 (0.0016) |
> | N3 (C) | -0.0085 (0.0012) | -0.0080 (0.0013) | -0.0075 (0.0011) | -0.0090 (0.0014) | -0.0103 (0.0013) |
> | N4 | -0.0105 (0.0013) | -0.0093 (0.0014) | -0.0094 (0.0013) | -0.0112 (0.0015) | -0.0127 (0.0016) |
>
> * Playstyle Distance (mix): consistent count = 2
>
> | Speed70N2 | Speed60 | Speed65 | Speed70 | Speed75 | Speed80 |
> | -------- | -------- | -------- | -------- | -------- | -------- |
> | N0 | -0.0033 (0.0006) | -0.0029 (0.0006) | -0.0029 (0.0006) | -0.0033 (0.0005) | -0.0046 (0.0007) |
> | N1 | -0.0035 (0.0006) | -0.0032 (0.0007) | -0.0034 (0.0007) | -0.0036 (0.0007) | -0.0046 (0.0006) |
> | N2 (C) | -0.0038 (0.0005) | -0.0035 (0.0008) | -0.0031 (0.0007) | -0.0037 (0.0006) | -0.0047 (0.0009) |
> | N3 (C) | -0.0039 (0.0006) | -0.0036 (0.0006) | -0.0034 (0.0005) | -0.0040 (0.0007) | -0.0047 (0.0007) |
> | N4 | -0.0046 (0.0006) | -0.0037 (0.0005) | -0.0040 (0.0006) | -0.0046 (0.0006) | -0.0060 (0.0009) |
>
> * Playstyle Intersection Similarity (mix): consistent count = 2
>
> | Speed70N2 | Speed60 | Speed65 | Speed70 | Speed75 | Speed80 |
> | -------- | -------- | -------- | -------- | -------- | -------- |
> | N0 (C) | 0.6774 (0.0241) | 0.6944 (0.0206) | 0.6995 (0.0233) | 0.6625 (0.0246) | 0.6286 (0.0265) |
> | N1 | 0.6295 (0.0252) | 0.6609 (0.0259) | 0.6603 (0.0247) | 0.6456 (0.0253) | 0.6031 (0.0281) |
> | N2 (C) | 0.5962 (0.0323) | 0.6376 (0.0308) | 0.7100 (0.0231) | 0.6086 (0.0291) | 0.6034 (0.0273) |
> | N3 | 0.5891 (0.0272) | 0.5844 (0.0277) | 0.5999 (0.0261) | 0.5764 (0.0297) | 0.5486 (0.0256) |
> | N4 | 0.5129 (0.0306) | 0.5664 (0.0274) | 0.5503 (0.0278) | 0.5135 (0.0297) | 0.4841 (0.0253) |
>
> * Playstyle Jaccard Index (mix): consistent count = 1
>
> | Speed70N2 | Speed60 | Speed65 | Speed70 | Speed75 | Speed80 |
> | -------- | -------- | -------- | -------- | -------- | -------- |
> | N0 | 0.0823 (0.0046) | 0.0829 (0.0047) | 0.0827 (0.0046) | 0.0844 (0.0043) | 0.0831 (0.0051) |
> | N1 | 0.0836 (0.0050) | 0.0820 (0.0048) | 0.0846 (0.0043) | 0.0850 (0.0045) | 0.0855 (0.0050) |
> | N2 (C) | 0.0821 (0.0050) | 0.0825 (0.0043) | 0.0930 (0.0049) | 0.0845 (0.0042) | 0.0842 (0.0045) |
> | N3 | 0.0806 (0.0042) | 0.0827 (0.0044) | 0.0795 (0.0049) | 0.0839 (0.0043) | 0.0780 (0.0045) |
> | N4 | 0.0802 (0.0046) | 0.0792 (0.0046) | 0.0799 (0.0048) | 0.0782 (0.0047) | 0.0804 (0.0050) |
>
> * Playstyle Similarity (mix): consistent count = 3
>
> | Speed70N2 | Speed60 | Speed65 | Speed70 | Speed75 | Speed80 |
> | -------- | -------- | -------- | -------- | -------- | -------- |
> | N0 (C) | 0.0554 (0.0040) | 0.0573 (0.0039) | 0.0579 (0.0043) | 0.0568 (0.0034) | 0.0522 (0.0037) |
> | N1 (C) | 0.0527 (0.0036) | 0.0555 (0.0041) | 0.0560 (0.0035) | 0.0549 (0.0038) | 0.0518 (0.0037) |
> | N2 (C) | 0.0486 (0.0038) | 0.0522 (0.004) | 0.0652 (0.0043) | 0.0512 (0.0035) | 0.0498 (0.0035) |
> | N3 | 0.0486 (0.0033) | 0.0487 (0.0035) | 0.0479 (0.0037) | 0.0485 (0.0038) | 0.0438 (0.0030) |
> | N4 | 0.0417 (0.0033) | 0.0452 (0.0033) | 0.0444 (0.0037) | 0.0407 (0.0035) | 0.0399 (0.0035) |

---

> > ### Comment · Reviewer_MAaz · 2023-11-15
> >
> > I'm pretty confused by these tables. Could the authors please put this in an appendix with the interpretation and a visualization that can help understand the consistency outcomes?

---

> > > ### Author Response · Authors · 2023-11-15
> > >
> > > Thank you for asking this question. We plan to use both textual interpretation and visualization to clarify these tables in the camera-ready version (if accepted). Additionally, we will provide further explanations in an appended comment should any other reviewer require more clarity.

---

### Author Response · Authors · 2023-11-16

Hello Reviewers,

We are conducting two additional experiments in response to Reviewer MAaz's request for more game datasets that were not evaluated in our paper or the baseline paper.

The first experiment involves using the RGSK training dataset provided in the official release of Lin et al., (2021), which was used for training the discrete representation. Since the training was focused on learning discrete representation and not encoding playstyle, we are using this as an additional dataset for playstyle analysis, even though it may not maintain a stable playstyle. This dataset contains 29 game episodes, and we will perform playstyle classification tasks on these results.

The second set of experiments will utilize new datasets we collected for diversity experiments (Section A.4). These datasets include 4 levels of stochasticity in a DRL policy, with 100 trajectories collected for each level, totaling 400 trajectories for each Atari game. We plan to use the first 20 trajectories from each level as a reference set and perform policy identification tasks on the remaining 320 trajectories (4*80) to calculate accuracy. These experiments will be completed soon in these days.

Additionally, while implementing these new experiments, we discovered an error in the implementation of our Playstyle Jaccard Index for RGSK and Atari games (it was a copy of Playstyle Similarity), though the TORCS version was correct. This bug can be found in the Supplementary Material:
ICLR2024Supplementary117\platform\cgi_drl\problem\atari\playstyle_distance\solver_iou.py
ICLR2024Supplementary117\platform\cgi_drl\problem\atari\playstyle_distance\mix_solver_iou.py
ICLR2024Supplementary117\platform\cgi_drl\problem\rgsk\playstyle_distance\solver_iou.py

We have fixed this bug and rerun the corresponding experiments. The manuscript PDF is updated, along with the relevant figures and descriptions. The updated results show that the Playstyle Jaccard Index performs slightly worse than Playstyle Similarity. We have highlighted the text changes in red in Sections A.3.1, A.3.2, and A.3.3 for clarity.
The updated figures, which include only updates to the Playstyle Jaccard Index (the red curve), are:
Figure 4 (b)(c)
Figure 11 (a)(b)(c)(d)(e)(f)(g)
Figure 12 (a)(b)(c)(d)(e)(f)(g)(h)
Figure 13 (b)

---

> ### Author Response · Authors · 2023-11-20
> **New Results (1/2)**
>
> Hello Reviewers,
>
> The new experiments involving additional gaming datasets have been completed. These datasets are unstable and include noise, therefore, we also compared the expected version of playstyle metrics (measures) in addition to the uniform version presented in our main paper. If applicable, we will add a new appendix section (A.5) in the camera-ready version to describe these variants. The primary difference lies in how each discrete state is weighted. For the baseline (Playstyle Distance), expected distributions according to observations are used. In contrast, for the metrics introduced after Section 3.2 in our paper, we apply a uniform distribution to weight each discrete state. This difference is not significant in stable or clear playstyles, but in unstable playstyles or with large action noises, the expected version sometimes yields better results.
>
> Due to the unsuitability of OpenReview for long mathematical equations, we have added the extra metrics tested in the new experiments to Section A.5 of the updated manuscript for review.
>
>
> ## RGSK 29 Trajectories
> We used the 29 trajectories from the RGSK racing game, which were utilized in HSD training, as the basis for the playstyle identification task. Each trajectory is considered as a distinct playstyle, and we performed the same task as in our main paper for RGSK. The comparison sample size for each dataset is 1024 observation-action pairs.
>
> |Accuracy | Playstyle Distance ($2^{20}$) | Playstyle Distance (Multiscale)  | Playstyle Intersection Similarity | Playstyle Jaccrad Index | Playstyle Similarity |
> | -------- | -------- | -------- | -------- | -------- | -------- |
> | Uniform | 41.84 | 43.13 | 41.57 |  99.82 | 99.83 |
> | Expected | 26.40 | 85.29 | 86.82 | 99.79 | 92.99 |
>
> These results indicate that the choice between uniform or expected versions significantly impacts accuracy, but the Jaccard Index is the most critical factor in this task.

---

> ### Author Response · Authors · 2023-11-20
> **New Results (2/2)**
>
> ## Atari 80-20 Trajectories Identification
> We used four levels of stochasticity from a single DRL model in our datasets, each consisting of 100 trajectories. We selected the first 20 trajectories of each level as the reference datasets for the four playstyles (stochasticity levels), and assessed the accuracy of predictions for the remaining 320 trajectories (4*80). Each trajectory is treated as a target dataset.
>
> ### Asterix
> | Accuracy | Playstyle Distance ($2^{20}$) | Playstyle Distance (Multiscale)  | Playstyle Intersection Similarity | Playstyle Jaccrad Index | Playstyle Similarity |
> | -------- | -------- | -------- | -------- | -------- | -------- |
> | Uniform | 53.02 | 46.77 | 39.27 |  61.77 | 54.68 |
> | Expected | 66.77 | 62.50 | 57.71 | 64.38 | 58.65 |
> ### Breakout
> | Accuracy | Playstyle Distance ($2^{20}$) | Playstyle Distance (Multiscale)  | Playstyle Intersection Similarity | Playstyle Jaccrad Index | Playstyle Similarity |
> | -------- | -------- | -------- | -------- | -------- | -------- |
> | Uniform | 54.58 | 52.92 | 44.79 | 51.25 | 51.25 |
> | Expected | 55.42 |  53.64 | 54.48 | 34.58 | 61.46 |
> ### MsPacman
> | Accuracy | Playstyle Distance ($2^{20}$) | Playstyle Distance (Multiscale)  | Playstyle Intersection Similarity | Playstyle Jaccrad Index | Playstyle Similarity |
> | -------- | -------- | -------- | -------- | -------- | -------- |
> | Uniform | 54.48 | 50.73 | 44.07 |  32.29 | 32.50 |
> | Expected | 65.94 |  55.94 | 53.96 | 58.65 | 58.33 |
> ### Pong
> | Accuracy | Playstyle Distance ($2^{20}$) | Playstyle Distance (Multiscale)  | Playstyle Intersection Similarity | Playstyle Jaccrad Index | Playstyle Similarity |
> | -------- | -------- | -------- | -------- | -------- | -------- |
> | Uniform | 68.23 | 64.90 | 52.40 |  30.00 | 30.00 |
> | Expected | 73.65 |  86.98 | 90.31 | 84.90 | 92.29 |
> ### Qbert
> | Accuracy | Playstyle Distance ($2^{20}$) | Playstyle Distance (Multiscale)  | Playstyle Intersection Similarity | Playstyle Jaccrad Index | Playstyle Similarity |
> | -------- | -------- | -------- | -------- | -------- | -------- |
> | Uniform | 47.08 | 43.02 | 30.73 |  25.83 | 26.15 |
> | Expected | 54.27 |  50.62 | 49.79 | 51.77 | 51.56 |
> ### Seaquest
> | Accuracy | Playstyle Distance ($2^{20}$) | Playstyle Distance (Multiscale)  | Playstyle Intersection Similarity | Playstyle Jaccrad Index | Playstyle Similarity |
> | -------- | -------- | -------- | -------- | -------- | -------- |
> | Uniform | 52.92 | 52.71 | 47.92 |  33.86 | 50.10 |
> | Expected | 53.23 |  53.64 | 53.54 | 53.96 | 53.64 |
> ### SpaceInvaders
> | Accuracy | Playstyle Distance ($2^{20}$) | Playstyle Distance (Multiscale)  | Playstyle Intersection Similarity | Playstyle Jaccrad Index | Playstyle Similarity |
> | -------- | -------- | -------- | -------- | -------- | -------- |
> | Uniform | 38.44 | 44.38 | 48.33 |  25.00 | 40.11 |
> | Expected | 60.52 |  55.73 | 49.79 | 45.73 | 52.29 |
>
> ### Summary of Atari Results
> These results demonstrate that the Jaccard Index can sometimes have negative effects, highlighting the importance of high-quality discrete representation. We encourage the ICLR community to delve deeper into discrete representation for playstyle-related topics.

---

### Meta-Review · Area_Chair_DAV7 · 2023-12-13

**Metareview:**

An improvement on a previously proposed Playstyle Distance metric is proposed. This metric is meant to be used to identify and disambiguate playing styles.

The problem is not very well studied, with only one previous approach identified. This is good from an originality perspective. It is also bad from the perspective of proving good performance, as there is not much to compare with. More importantly, it is bad from the perspective of showing that this is a problem people care about.

Ultimately, neither I nor the reviewers are convinced that the problem as stated is very important. This is in my opinion the main reason to not accept it. A secondary reason is that the technical contribution does not appear to be very deep. It is possible that some future version of this paper could justify the problem and the solution strategy better.

**Justification For Why Not Higher Score:**

None of the reviewers is interested in accepting the paper.

**Justification For Why Not Lower Score:**

N/A

---

### Decision · Program_Chairs · 2024-01-16

Reject